# Thermodynamic Analysis of Chemical Hydrogen Storage: Energetics of Liquid Organic Hydrogen Carrier Systems Based on Methyl-Substituted Indoles

**DOI:** 10.3390/ma16072924

**Published:** 2023-04-06

**Authors:** Sergey V. Vostrikov, Artemiy A. Samarov, Vladimir V. Turovtsev, Peter Wasserscheid, Karsten Müller, Sergey P. Verevkin

**Affiliations:** 1Chemical-Technological Department, Samara State Technical University, 443100 Samara, Russia; 2Department of Chemical Thermodynamics and Kinetics, Saint Petersburg State University, 198504 Saint Petersburg, Russia; samarov@yandex.ru; 3Department of Physics, Tver State Medical University, 170100 Tver, Russia; turtsma@mail.ru; 4Institute of Chemical Reaction Engineering, Friedrich-Alexander-Universität Erlangen-Nürnberg, Egerlandstr. 3, 91058 Erlangen, Germany; peter.wasserscheid@fau.de; 5Forschungszentrum Jülich GmbH, Helmholtz Institute Erlangen-Nürnberg for Renewable Energy (IEK-11), Egerlandstr. 3, 91058 Erlangen, Germany; karsten.mueller@uni-rostock.de; 6Institute of Technical Thermodynamics, University of Rostock, Albert-Einstein Str. 2, 18059 Rostock, Germany; 7Competence Centre CALOR of the Department Life, Light & Matter, Faculty of Interdisciplinary Research, University of Rostock, 18059 Rostock, Germany; 8Department of Physical Chemistry, Kazan Federal University, 420008 Kazan, Russia

**Keywords:** vapor pressure, enthalpy of vaporization, enthalpy of formation, structure–property relationships, quantum chemical calculations

## Abstract

Liquid organic hydrogen carriers can store hydrogen in a safe and dense form through covalent bonds. Hydrogen uptake and release are realized by catalytic hydrogenation and dehydrogenation, respectively. Indoles have been demonstrated to be interesting candidates for this task. The enthalpy of reaction is a crucial parameter in this regard as it determines not only the heat demand for hydrogen release, but also the reaction equilibrium at given conditions. In this work, a combination of experimental measurements, quantum chemical methods and a group-additivity approach has been applied to obtain a consistent dataset on the enthalpies of formation of different methylated indole derivatives and their hydrogenated counterparts. The results show a namable influence of the number and position of methyl groups on the enthalpy of reaction. The enthalpy of reaction of the overall hydrogenation reaction varies in the range of up to 18.2 kJ·mol^−1^ (corresponding to 4.6 kJ·mol(H_2_)^−1^). The widest range of enthalpy of reaction data for different methyl indoles has been observed for the last step (hydrogenation for the last double bond in the five-membered ring). Here a difference of up to 7.3 kJ·mol(H_2_)^−1^ between the highest and the lowest value was found.

## 1. Introduction

Chemical hydrogen storage and release processes are essential prerequisites for the implementation of a sustainable energy system [1]. The last decade has seen rapid growth in research activities on hydrogen storage materials. Reversible hydrogenation/dehydrogenation of aromatic compounds suitable as liquid organic hydrogen carriers (LOHCs) is considered a promising alternative to conventional hydrogen storage technologies. There are different types of hydrogen carriers that can be utilized as LOHCs. Sometimes substances that are irreversibly decomposed to hydrogen and carbon dioxide (such as formic acid or methanol) are included in the term [2]. However, the most promising materials are those that can be applied in multiple cycles. Chemical hydrogen storage with these LOHC systems is usually achieved by catalytic hydrogenation of a material containing unsaturated or aromatic molecules (hydrogen uptake), which are then reversibly dehydrogenated on a catalyst, releasing the hydrogen that can be used subsequently [3,4]. The main advantage of LOHC-based hydrogen storage compared to, e.g., solid carrier materials is the fact that the hydrogen is stored in a liquid material whose physical properties are beneficial for handling in a fuel-like manner. For instance, the liquid nature of the material enables the pumping of the carrier and control of hydrogen release by removing the catalyst. Furthermore, the kinetics of hydrogen uptake and release are much better than those in metal hydrides. A slight disadvantage is the fact that hydrogen needs purification after release as small amounts of the carrier are evaporated.

Indole derivatives are considered as promising liquid organic hydrogen carriers for on-board hydrogen storage applications [5]. A property that makes indoles particularly attractive compared to other materials is the rather low enthalpy of reaction for hydrogen release. The indole derivatives are among the most common and important heterocycles in nature. The continuous development of routes to indoles has been a central theme in organic synthesis over the last century, which is commensurate with their importance [6]. Indole and its derivatives can be synthesized by a variety of conventional methods [7,8,9]. Moreover, the indole derivatives can also be biosynthesized [10].

The kinetics of hydrogenation/dehydrogenation reactions of methyl-indoles have been intensively studied in the recent past [11,12,13]. The kinetics of hydrogenation of 2-methyl-indole were studied over the Ru/Al_2_O_3_ (5 wt%) catalyst in the temperature range of 120–170 °C at a hydrogen pressure of 7 MPa. Reversible dehydrogenation was achieved with the same catalyst in 4 h at 190 °C [11].

The hydrogenation of 1-methylindole and the dehydrogenation of octahydro-1-methylindole were investigated over a 5 wt% Ru/Al_2_O_3_ catalyst. Hydrogenation with nearly 100% conversion and selectivity was easily achieved at 130 °C and 6.0 MPa. The successful dehydrogenation was performed in the temperature range of 160–190 °C [12].

Full hydrogenation of 1,2-dimethyl-indole can be realized at 140 °C and 7 MPa in 60 min over a 5 wt% Ru/Al_2_O_3_ catalyst. The stored hydrogen can be completely released via perhydro-1,2-dimethyl-indole dehydrogenation at 200 °C and 101 kPa within 60 min over the same catalyst [13].

For 2,3-dimethylindole, complete hydrogenation was achieved over 5 wt% Ru/Al_2_O_3_ at 190 °C and 7 MPa in 4 h. Dehydrogenation of perhydro-2,3-dimethylindole was successfully performed over 5 wt% Pd/Al_2_O_3_ at 180–210 °C and 101 kPa [14]

All these examples show the principle feasibility of hydrogenation/dehydrogenation cycles using methyl indole derivatives. However, the optimization of the technological processes requires extended thermodynamic data on LOHC systems consisting of both counterparts (hydrogen-lean material (indoles) and hydrogen-rich material (perhydro-hydrogenated indoles)), which are the subject of the present work. The present work continues the series of our earlier thermodynamic work [3,15] on indole derivatives. In contrast to previous studies, which focused on intermediates and the general effects of substitution [4], this work focuses on the reversible hydrogenation/dehydrogenation reactions in the dimethylindole-based LOHC systems (see Figure 1).

The energetics of these reactions are essential for chemical engineering calculations and the optimization of the heat streams of technological processes. The standard molar enthalpies of chemical reactions, $\Delta_{r}H_{m}^{o}$, are calculated from the enthalpies of formation of reactants and products according to Hess’s law, e.g., for the complete hydrogenation reaction of a methyl indole:(1)$$\Delta_{r}H_{m}^{o}=\Delta_{f}H_{m}^{o}(\mathrm{HR})-\Delta_{f}H_{m}^{o}(\mathrm{HL})-\Delta_{f}H_{m}^{o}(H_{2})=\Delta_{f}H_{m}^{o}\left(\mathrm{HR})-\Delta_{f}H_{m}^{o}(\mathrm{HL}\right)$$
where $\Delta_{f}H_{m}^{o}$(HR) and $\Delta_{f}H_{m}^{o}$(HL) are the standard molar enthalpies of formation of the hydrogen-rich and hydrogen-lean counterparts of the LOHC system (for example 3-methyl-(H8)-indole as HR counterpart and 3-methyl-indole as HL counterpart).

The catalytic process of reversible hydrogen storage and release is designed in the *liquid phase* at elevated temperatures and pressures; therefore, the standard molar enthalpies of formation, $\Delta_{f}H_{m}^{o}$(liq), of the LOHC counterparts in the *liquid phase* are needed. However, the purely experimental determination of all necessary thermodynamic data is thwarted by technical complications. In a number of our recent works [3,15], however, we have shown that a reasonable combination of experimental, empirical and quantum chemical calculations makes it possible to reduce the experimental effort without losing the reliability of the $\Delta_{f}H_{m}^{o}$(liq)-values.

The algorithm to derive the $\Delta_{f}H_{m}^{o}$(liq)-values consists of a few steps and is based on the general equations relating the thermochemical properties:(2)$$\Delta_{f}H_{m}^{o}(\mathrm{liq})=\Delta_{f}H_{m}^{o}(g)-\Delta_{l}^{g}H_{m}^{o}$$
(3)$$\Delta_{f}H_{m}^{o}(\mathrm{liq})=\Delta_{f}H_{m}^{o}(g)-(\Delta_{\mathrm{cr}}^{g}H_{m}^{o}-\Delta_{\mathrm{cr}}^{l}H_{m}^{o})$$
where the standard molar vaporization enthalpies, $\Delta_{l}^{g}H_{m}^{o}$, the standard molar sublimation enthalpies, $\Delta_{\mathrm{cr}}^{g}H_{m}^{o}$, and the standard molar fusion enthalpies, $\Delta_{\mathrm{cr}}^{l}H_{m}^{o}$, are usually measured by different experimental methods [16]. In thermochemistry, it is common to adjust all enthalpies involved in Equations (1)–(3) to an arbitrary but common reference temperature. In this work, a reference temperature of *T* = 298.15 K was chosen.

*Step I*: In the first step, high-level quantum chemical (QC) methods of the G*-family (e.g., G3MP2 [17], G4 [18]) and the CBS-APNO [19] method are used to derive the gas-phase enthalpies of formation, $\Delta_{f}H_{m}^{o}$(g, 298.15 K), for both HL and HR counterparts of the LOHC systems. In our experience [3,15], the results of these methods agree well with the available experimental gas-phase enthalpies of formations.

*Step II*: The vaporization enthalpies, $\Delta_{l}^{g}H_{m}^{o}$(298.15 K) required for the calculations according to Equation (2), are collected from literature and validated with complementary measurements.

The sublimation enthalpies, $\Delta_{\mathrm{cr}}^{g}H_{m}^{o}$(298.15 K) required for the calculations according to Equation (3), are collected from literature and also validated with complementary measurements. The fusion enthalpies, $\Delta_{\mathrm{cr}}^{l}H_{m}^{o}$, which are also needed for the calculation of the enthalpies of formation in the *liquid phase* according to Equation (3), can easily be measured by differential scanning calorimetry (DSC).

*Step III*: The vaporization enthalpies, $\Delta_{l}^{g}H_{m}^{o}$(298.15 K) required for the calculations according to Equation (2), are often missing, especially for the alicyclic, hydrogen-rich materials. In such cases, $\Delta_{l}^{g}H_{m}^{o}$(298.15 K)-values can be obtained from correlation with measurable physico-chemical properties (e.g., normal boiling temperatures or gas-chromatographic retention indices). These different types of correlations not only provide the missing values, but also cross-link the vaporization enthalpies of HL and HR materials to the network of reliable data and provide confidence in the evaluated numerical value. For such cases, we have also developed a “centerpiece” approach that is suitable for a reliable appraisal of the required thermodynamic data and is based on the principles of group additivity.

*Step IV*: The target enthalpies of formation of the HL and HR materials in *liquid phase* are derived according to Equations (2) and (3) and used in Equation (1) to estimate the hydrogenation/dehydrogenation enthalpies of the LOHC systems and to analyze how the structural features affect the energetics of this process.

The “step-by-step” evaluation of the thermodynamic properties of the HR and HL materials leading to the energetics of hydrogenation/dehydrogenation reactions with methyl-indole derivatives and the analysis of these results are the focus of this work.

## 2. Methods

### 2.1. Materials

Samples of 3-methyl-indole and 1,2-dimethyl-indole were of commercial origin (see Appendix A) with purities of 0.99 mass fraction as given in the specification. Prior to the experiment, the samples were purified using fractional vacuum sublimation. Purities were determined using a gas chromatograph equipped with a flame ionization detector and a capillary column HP-5 (stationary phase crosslinked 5% PH ME silicone, column length of 30 m, inside diameter of 0.32 mm, film thickness of 0.25 μm). The analysis was performed with the temperature program *T* = 353 K for 30 s followed by a heating rate of 10 K·min^−1^ to *T* = 523 K. No contamination (greater than the mass fraction 0.0009) could be detected in the samples used for the thermochemical measurements.

### 2.2. Theoretical and Experimental Thermochemical Methods

The theoretical gas-phase enthalpies of methyl-indoles were calculated using the composite QC methods [16,17,18,19] from the Gaussian 16 suitcase software [20]. The *H*_298_-values were finally converted to the $\Delta_{f}H_{m}^{o}$(g, 298.15 K)_theor_ values and discussed. For quantum chemical calculations, the most stable conformer of each compound was selected. The well-established assumption “rigid rotator–harmonic oscillator” was used for the quantum chemical calculations. Details on the calculation methods have been reported elsewhere [21].

The transpiration method [22] was applied to measure the vapor pressures of 3-methyl-indole and 1,2-dimethyl-indole at different temperatures. The standard molar enthalpies of sublimation, $\Delta_{\mathrm{cr}}^{g}H_{m}^{o}$, for 3-methyl-indole and 1,2-dimethyl-indole, as well as the standard molar enthalpy of vaporization, $\Delta_{l}^{g}H_{m}^{o}$, of 1,2-dimethyl-indole were derived from the temperature dependencies of the vapor pressures. The details of the experimental technique are given in the Electronic Support Information (ESI).

## 3. Results

### 3.1. Step I: Gas-Phase Standard Molar Enthalpies of Formation: Theory and Experiment

Quantum chemical methods have become a valuable tool to obtain the *theoretical* $\Delta_{f}H_{m}^{o}$(g, 298.15 K)-values with “chemical accuracy” (conventionally at the level of 4–5 kJ·mol^−1^) [23]. In our recent work [3], we tested the G4 composite method by comparing the results with experimental data obtained by high-precision combustion calorimetry and vapor pressure measurements (see Table 1).

As can be seen from this table, the deviation between the G4 method and experimental results is no more than 2–3 kJ·mol^−1^, which is even better than the claimed “chemical accuracy”. However, in this work, the G3MP2 and CBS-APNO methods were tested additionally to estimate the *theoretical* $\Delta_{f}H_{m}^{o}$(g, 298.15 K)-value of 3-methyl-indole as an example. The simultaneous use of several methods helps to avoid possible systematic errors in the calculations. Stable conformers of 3-methyl-indole were found using a CREST (Conformer-Rotamer Ensemble Sampling Tool) computer code [26] and optimized using the B3LYP/6-31g(d,p) method [27]. Since the structures of the 3-methyl indole conformers are flat, the energetic differences do not exceed 1 kJ·mol^−1^. Therefore, the high-level calculations were performed only for the most stable conformer. The energy *E*_0_ and the enthalpy *H*_298_ of the most stable conformer were finally calculated by using the high-level quantum chemical methods. The *H*_298_-values were converted to the standard molar enthalpies of formation $\Delta_{f}H_{m}^{o}$(g, 298.15 K)_AT_ by using the atomization (AT) reaction:
*C*_a_*H*_b_*N*_c_ = a × *C* + b × *H* + c × *N*
(4)


In addition, the *H*_298_ enthalpies of 3-methyl-indole were converted to the enthalpies of formation using the enthalpies of the homodesmotic reactions shown in Figure 2.

To derive the *theoretical* $\Delta_{f}H_{m}^{o}$(g, 298.15 K), the experimental enthalpies of formation of the homodesmotic reaction participants were used. They are compiled in Appendix A. The quantum chemical reaction enthalpies are given in Appendix A. The resulting $\Delta_{f}H_{m}^{o}$(g, 298.15 K)-values for each reaction are given in Table 2, as calculated by the methods G4, G3MP2 and CBS-APNO.

As shown in Table 2, the theoretical enthalpies of formation of 3-methyl-indole calculated by G4, G3MP2 and CBS-APNO using atomization and homodesmotic reactions are quite close to the experimental value $\Delta_{f}H_{m}^{o}$(g, 298.15 K)_exp_ = 128.7 ± 2.4 kJ·mol^−1^ (see Table 1). In particular, the results of the G4 method are remarkably consistent, regardless of the type of reaction used to derive the gas-phase enthalpy of formation. Furthermore, the results of the G4 calculations for the series of indole derivatives compiled in Table 1 show very good agreement with the experimental data, even when the simplest atomization procedure, according to Equation (4), was used. In order to reduce the computational effort for the indole derivatives of interest in this study (see Figure 1), the calculations of the theoretical gas-phase enthalpies of formation were therefore only carried out using the G4 method and the atomization procedure. The structures of the most stable conformers and the resulting $\Delta_{f}H_{m}^{o}$(g, 298.15 K)_G4_-values for methyl- and dimethyl-indole derivatives are listed in Table 3.

The gas-phase enthalpies of formation of methylindoles calculated in Table 3 can now be substituted into Equations (2) and (3) to derive the hydrogenation reaction enthalpies for the LOHC systems based on methylindole derivatives.

### 3.2. Step II: Vaporization Enthalpies of Indole Derivatives

#### 3.2.1. Experimental Vapor Pressures

3-Methyl-indole and 1,2-dimethyl-indole are solids at room temperature with respective melting points of 368.0 K and 330.7 K, as given in Appendix A. Moreover, for 3-methyl-indole, a solid–solid phase transition was observed at 316.8 K [4]. Therefore, for the correct interpretation of the phase transition data at the reference temperature *T* = 298.15 K (see Appendix A), the sublimation enthalpy of 3-methylindole was measured below the phase transition temperature 316.8 K (see Appendix A). The experimental dependences of the vapor pressures, *p_i_*, on temperature measured in this work for 3-methyl-indole and 1,2-dimethyl-indole (see Table 4) were correlated by the following equation [22]:(5)$$R\times \ln(p_{i}/p_{ref})=a+\frac{b}{T}+\Delta_{\mathrm{cr},l}^{g}C_{p,m}^{o}\times \ln\left(\frac{T}{T_{0}}\right),$$
where $\Delta_{\mathrm{cr},l}^{g}C_{p,m}^{o}$ is the difference between the molar heat capacities of the gas and the crystal (or liquid) phases (see Appendix A), *a* and *b* are adjustable parameters, R = 8.31446 J·K^−1^·mol^−1^ is the molar gas constant, and the reference pressure $p_{ref}=1\;\mathrm{Pa}.$ The arbitrary temperature *T*_0_ given in Equation (5) was chosen to be *T*_0_ = 298.15 K. The results of the vapor pressure measurements using the transpiration method are shown in Table 4.

Experimental vapor pressures have been used to obtain the enthalpies of sublimation/vaporization using the following equation:(6)$$\Delta_{\mathrm{cr},l}^{g}H_{m}^{o}\left(T\right)=-b+\Delta_{\mathrm{cr},l}^{g}C_{p,m}^{o}\times T$$

Experimental vapor pressures temperature dependences were also used to derive the sublimation/vaporization entropies at temperatures *T* by using the following equation:(7)$$\Delta_{\mathrm{cr},l}^{g}S_{m}^{o}\left(T\right)=\Delta_{\mathrm{cr},l}^{g}H_{m}^{o}/T+R\times \ln\left(p_{i}/p^{o}\right)$$
with $p^{o}$ = 0.1 MPa. Coefficients *a* and *b* of Equation (5), $\Delta_{\mathrm{cr},l}^{g}H_{m}^{o}$(*T*) and $\Delta_{\mathrm{cr},l}^{g}S_{m}^{o}$(*T*) values are collected in Table 4. According to general practice, all thermochemical quantities must be presented at the reference temperature *T* = 298.15 K. Details of the temperature adjustment are given in Appendix A. The resulting sublimation/vaporization enthalpies at the reference temperature *T* = 298.15 K are given in Table 5, column 5.

The literature on vapor pressures of methyl indoles is sparse. The two sets found in the compilations [29,30] were approximated according to Equation (5), and the vaporization enthalpies are given in Table 5 for comparison. The uncertainties in the temperature adjustment of these enthalpies to the reference temperature *T* = 298.15 K were estimated to account for 20% of the total adjustment.

#### 3.2.2. Evaluation of the Vaporization Enthalpies by Consistency of Phase Transitions Solid–Gas, Liquid–Gas and Solid–Liquid

For 3-methylindole, a solid–solid phase transition from phase II to phase I was observed at Δ*T*_trs_ = 316.8 K [3]. The energetics of phase transition $\Delta_{\mathrm{phase}\;\mathrm{II}}^{\mathrm{phase}\;I}H_{m}^{o}$ = 2.7 ± 0.6 kJ·mol^−1^ was measured at Δ*T*_trs_ [3]. The enthalpy of fusion of the phase I of 3-methyl-indole $\Delta_{\mathrm{cr}}^{l}H_{m}^{o}$(*T*_fus_) = 11.6 ± 0.5 kJ·mol^−1^ (see Appendix A) was measured in our previous work [3]. The total energetics of phase transitions below melting point for this compound was calculated as the sum of $\Delta_{\mathrm{phase}\;\mathrm{II}}^{\mathrm{phase}\;I}H_{m}^{o}$ and $\Delta_{\mathrm{cr}}^{l}H_{m}^{o}$(*T*_fus_) as recommended by Acree and Chickos [31]. The latter sum was adjusted to the reference temperature *T* = 298.15 K with help of Equation (8) [31]:(8)$$\Delta_{\mathrm{cr}}^{l}H_{m}^{o}(298.15\;K)/(J\cdot {\mathrm{mol}}^{-1})=\Delta_{\mathrm{cr}}^{l}H_{m}^{o}(T_{\mathrm{fus}}/K)-(\Delta_{\mathrm{cr}}^{g}C_{p,m}^{o}-\Delta_{l}^{g}C_{p,m}^{o})\times [(T_{\mathrm{fus}}/K)-298.15\;K]$$
where $\Delta_{\mathrm{cr}}^{g}C_{p,m}^{o}$ and $\Delta_{l}^{g}C_{p,m}^{o}$ are given in Appendix A. With this adjustment, the molar enthalpy of fusion $\Delta_{\mathrm{cr}}^{l}H_{m}^{o}$(298.15 K) = 11.4 ± 1.2 kJ·mol^−1^ (see Appendix A) of 3-methyl-indole was calculated. Uncertainty in the temperature adjustment of fusion enthalpy from *T*_fus_ to the reference temperature was estimated to account for 30% of the total adjustment [32]. The experimental vaporization enthalpy for 3-methyl-indole is missing in the literature. However, using the common thermochemical equation:(9)$$\Delta_{l}^{g}H_{m}^{o}(298.15\;K)=\Delta_{\mathrm{cr}}^{g}H_{m}^{o}(298.15\;K)-\Delta_{\mathrm{cr}}^{l}H_{m}^{o}(298.15\;K)=(81.4-11.4)=70.0\pm 1.3\;\mathrm{kJ}\cdot {\mathrm{mol}}^{-1}$$
it can be derived and inserted into Equations (2) and (3).

The vapor pressures of 1,2-dimethyl indole were studied for the first time. In this work, the vapor pressures of 1,2-dimethylindole were studied by the transpiration method at temperatures below and above its melting temperature. Results are given in Table 4. Values of sublimation enthalpy $\Delta_{\mathrm{cr}}^{g}H_{m}^{o}$(298.15 K) = 83.2 ± 0.7 kJ·mol^−1^ and vaporization enthalpy $\Delta_{l}^{g}H_{m}^{o}$(298.15 K) = 67.6 ± 0.7 kJ·mol^−1^ were measured. The fusion enthalpy of 1,2-dimethylindole $\Delta_{\mathrm{cr}}^{l}H_{m}^{o}$(298.15 K) = 13.3 ± 1.1 kJ·mol^−1^ is derived in Appendix A. The difference between the sublimation and fusion enthalpies, $\Delta_{l}^{g}H_{m}^{o}$(298.15 K) = 69.9 ± 1.3 kJ·mol^−1^, agrees with the transpiration enthalpy, $\Delta_{l}^{g}H_{m}^{o}$(298.15 K) = 67.6 ± 0.7 kJ·mol^−1^, which proves the consistency of the data measured in this work for 1,2-dimethylindole in the solid–gas, liquid–gas and solid–liquid phase transitions.

### 3.3. Step III: Determination of the Missing Vaporization Enthalpies

#### 3.3.1. Determination from Boiling Temperatures Available in the Literature at Different Pressures

Systematic vapor pressure measurements for dimethylindole derivatives are generally not found in the literature. In order to determine at least the general trends for these compounds, the experimental boiling temperatures available in the literature at different pressures [33] were collected in this work and approximated using Equation (5). The origin of these boiling points comes from the distillation of reaction mixtures after synthesis and not in special physico-chemical investigations. Usually, temperatures are given in the range of a few degrees, and pressures are measured with uncalibrated manometers. In our earlier work on methyl- and dimethyl-indoles, however, we have shown that reasonable trends can generally be derived even from such raw data [3]. The vaporization enthalpies, $\Delta_{l}^{g}H_{m}^{o}$(298.15 K), obtained in this way are referred to as boiling points (BPs) and are given in Table 5 for comparison with the results determined by other methods.

#### 3.3.2. Determination by Correlation with Retention Indices

It is well known that the $\Delta_{l}^{g}H_{m}^{o}$(298.15 K)-values correlate linearly with the gas-chromatographic retention indices in a series of structurally similar compounds. We have derived a reliable linear correlation between the enthalpies of vaporization, $\Delta_{l}^{g}H_{m}^{o}$(298.15 K)-values, of compounds with reliable experimental data and the Kovats indices, *J_x_*, available for these compounds in the literature [34] (see Table 6):
(10)$$\Delta_{l}^{g}H_{m}^{o}(298.15\;K)/(\mathrm{kJ}\cdot {\mathrm{mol}}^{-1})=1.8+0.0462\times J_{x}\;\mathrm{with}\;(R^{2}=0.997)$$

The enthalpies of vaporization of cyclic aliphatic and aromatic hydrocarbons derived from this correlation (see Table 6, column 4) agree well with those taken from the literature for the correlation. Table 6 shows that the differences between the experimental enthalpies of vaporization and the “empirical” values calculated according to Equation (10) are mostly less than 0.5 kJ·mol^−1^. Therefore, the uncertainties of the enthalpies of vaporization of 3-methyl-indoline and 3-methyl-(H8)-indole, estimated from the correlation of $\Delta_{l}^{g}H_{m}^{o}$(298.15 K) with Kovats indices, are evaluated with an uncertainty of ±1.0 kJ·mol^−1^.

Furthermore, reliable linear correlation between vaporization enthalpies $\Delta_{l}^{g}H_{m}^{o}$(298.15 K) and the gas-chromatographic Lee indices [38], *J_Lee_*, of methyl-indoles, methyl-quinolines and parent compounds (see Appendix A) have been derived.
(11)$$\Delta_{l}^{g}H_{m}^{o}(298.15\;K)/(\mathrm{kJ}\cdot {\mathrm{mol}}^{-1})=1.3+0.2712\times J_{\mathrm{Lee}}\;\mathrm{with}\;(R^{2}=0.999)$$

The vaporization enthalpy of 1,2-dimethyl-indole, $\Delta_{l}^{g}H_{m}^{o}$(298.15 K) = 67.6 ± 1.0 kJ·mol^−1^, derived from this correlation agrees perfectly with the value $\Delta_{l}^{g}H_{m}^{o}$(298.15 K) = 67.6 ± 0.7 kJ·mol^−1^ measured in this work. This good agreement can be seen as an additional validation of the experimental data measured in this work with the transpiration method (see Table 4).

#### 3.3.3. Determination by Correlation with Normal Boiling Temperatures *T_b_*

Another way to determine the missing enthalpies of vaporization is to correlate the enthalpies of vaporization, $\Delta_{l}^{g}H_{m}^{o}$(298.15 K), with the normal boiling temperatures [4]. The data available in the literature on the normal boiling temperatures, *T_b_*, of methyl-substituted indoles were correlated with the reliable $\Delta_{l}^{g}H_{m}^{o}$(298.15 K)-values available from the literature. For the set of indoles compiled in Table 7, the following linear correlation between the $\Delta_{l}^{g}H_{m}^{o}$(298.15 K)-values and their *T_b_* was obtained:(12)$$\Delta_{l}^{g}H_{m}^{o}(298.15\;K)/(\mathrm{kJ}\cdot {\mathrm{mol}}^{-1})=-91.7+0.2991\times T_{b}\;\mathrm{with}\;(R^{2}=0.988)\;$$

Table 7 shows that the differences between the experimental enthalpies of vaporization and the “empirical” values calculated according to Equation (12) are mostly less than 0.5 kJ·mol^−1^. Therefore, the uncertainty of the enthalpy of vaporization of 1,3-dimethylindole derived from this correlation was evaluated as ±1.0 kJ·mol^−1^.

#### 3.3.4. Assessment of the Missing Vaporization Enthalpies by the “Centerpiece” Approach

No data on the enthalpies of vaporization of 1,2-dimethyl-(H8)-indole and 1,3-dimethyl-(H8)-indole were found in the existing literature. The enthalpies of vaporization of 3-methyl-indoline, 2,3-dimethyl-indoline, 1,2-dimethyl-indoline and 1,3-dimethyl-indoline, which were determined from the boiling temperatures at different pressures (see Section 3.3.3), also require additional validation, as they were derived from data of uncertain quality. For these estimations, the “centerpiece” approach recently developed in our work was applied.

This approach is based on the well-established principles of group additivity (GA), which are basically used for the estimation of thermodynamic properties [39]. In the conventional way, the reliable experimental vaporization enthalpies for the widest possible range of molecules are divided into relatively small groups, like “LEGO bricks”. Using a matrix calculation, a precisely defined numerical contribution is attributed to each group. The prediction of the enthalpy of vaporization is then a simple construction of the desired molecule from the “bricks”, where the energetics of a molecule is assembled from the appropriate number and type of bricks. The GA method is simple and straightforward, but it is impractical for large molecules due to there being too many “building bricks”. To overcome this drawback, a general approach was developed to assess the vaporization enthalpies based on a so-called “centerpiece” molecule.

The idea behind the “centerpiece” approach is to start the prediction with a potentially large “core” molecule with a reliable $\Delta_{l}^{g}H_{m}^{o}$(298.15 K)-value that can generally mimic the structure of the molecule of interest. A visualization of the “centerpiece” approach to estimating $\Delta_{l}^{g}H_{m}^{o}$(298.15 K) of 3-methyl indoline from the reliable enthalpy of vaporization of indoline is shown in Figure 3.

The CH_3_-contribution estimated from reliable vaporization enthalpies of methyl-cyclopentane [40] and cyclopentane [40] can be considered universal in the context of this work and can be used for further estimates of the $\Delta_{l}^{g}H_{m}^{o}$(298.15 K)-values for 2,3-dimethyl-indoline, 1,3-dimethyl-indoline, 1,2-dimethyl-(H8)-indole and 1,3-dimethyl-(H8)-indole, as shown in Figure 4.

The “empirical” $\Delta_{l}^{g}H_{m}^{o}$(298.15 K)-values derived in Section 3.3.1, Section 3.3.2, Section 3.3.3 and Section 3.3.4 are compiled in Table 5 for comparison. It is obvious that they agree within their uncertainties for each compound. To gain more confidence in these results, the weighted average enthalpies for each indole derivative were estimated, and these values were recommended for thermochemical calculations according to Equation (2) to derive the liquid-phase enthalpies of formation for the LOHC systems based on methyl-indole derivatives.

### 3.4. Step IV: Liquid-Phase Standard Molar Enthalpies of Formation of HL and HR Materials

In this final step, the *gas-phase* enthalpies of formation, $\Delta_{f}H_{m}^{o}$(g, 298.15 K), of indole derivatives derived in step I (see Table 3) were used together with the vaporization enthalpies, $\Delta_{l}^{g}H_{m}^{o}$(298.15 K), evaluated in steps II and III. and these results were applied according to Equation (2) to obtain the *liquid-phase* $\Delta_{f}H_{m}^{o}$(liq, 298.15 K), given in Table 8.

With these $\Delta_{f}H_{m}^{o}$(liq, 298.15 K)-values for the HL and HR materials, the energetics of the hydrogenation/dehydrogenation reactions can now be calculated according to Equation (1). For a complete overview, our recent data on unsubstituted indole derivatives and 2-methyl indole derivatives [3] are also compiled in Table 8 and discussed.

## 4. Discussion

### 4.1. Thermodynamic Analysis of LOHC Systems (Combination of Experimental and Theoretical Methods)

#### 4.1.1. Energetics of Hydrogen Uptake and Release in the Liquid Phase

From a theoretical point of view, it is interesting to analyze the energetics of the stepwise indole hydrogenation as shown in Figure 5.

The partial hydrogenation of the double bond involved in the five-membered ring is represented by reaction R-I, and the consequent hydrogenation of all three double bonds in the six-membered ring is represented by reaction R-II.

From a practical point of view, the energetics of the complete hydrogenation of the aromatic system, represented by reaction R-III (see Figure 6), is crucial for optimizing the temperature management of a chemical reactor.

It should be noted that the reactions shown in Figure 5 and Figure 6 represent the generalization of the individual reactions of reversible hydrogenation/dehydrogenation in the methyl-indole-based LOHC systems shown in Figure 1.

The enthalpies of the hydrogenation/dehydrogenation reactions R-I, R-II and R-III were derived according to Hess’s law (see Equation (1)), using the *liquid-phase* standard molar enthalpies of formation of the reaction participants evaluated in Table 8. The *liquid-phase* enthalpies of the reversible dehydrogenation/hydrogenation reactions, $\Delta_{r}H_{m}^{o}$(liq), estimated according to Equation (1), are given in Table 9.

As can be seen from this table, the hydrogenation of the double bond in the five-membered ring (reaction R-I) has an enthalpy of reaction of about −33 kJ·mol^−1^ and does not differ significantly for all the structures considered in Table 9. The hydrogenation of three double bonds in the six-membered ring (reaction R-II) is more exothermal, with an enthalpy of reaction of about −173 kJ·mol^−1^ or relative to the hydrogen molecules −173/3 = −58 kJ·mol(H_2_)^−1^. On the other hand, this means that the dehydrogenation of the six-membered ring is more energetically demanding than the dehydrogenation of the five-membered ring.

It is interesting to compare the energetics of hydrogenation of double bonds in nitrogen-containing five-membered rings and in the corresponding carbocyclic aromatic compounds (see Figure 7).

Hydrogenation of the double bond in the nitrogen-containing five-membered ring of indole (-37.9 kJ·mol^−1^) has been found to be less than twice as exothermal than the corresponding bond in indene (−99.1 kJ·mol^−1^). Surprisingly, this ratio is significantly lower in the non-aromatic counterparts: the nitrogen-containing five-membered ring of 2-pyroline (−94.4 kJ·mol^−1^) and the corresponding carbocyclic five-membered ring of cyclopentene (−109.5 kJ·mol^−1^). The reason for this observation is most likely the conjugation of the π-electrons of benzene with the electronic orbitals of nitrogen. The significantly lower decrease in hydrogenation enthalpy was observed when all three double bonds in the six-membered ring of indole (−57.9 kJ·mol^−1^) were consistently hydrogenated compared to indane (−62.6 kJ·mol^−1^) or benzene (−68.5 kJ·mol^−1^) as shown in Figure 8.

Relating the reaction enthalpy to the amount of hydrogen captured or released (kJ·mol^−1^/_H2_) enables a comparison of the enthalpy values of LOHC systems with different stoichiometries and is therefore important from a practical point of view. These $\Delta_{r}H_{m}^{o}$(liq)H_2_-values are given in Table 9, column 5. In these units, the hydrogenation enthalpies of mono-methylated and di-methylated indoles range between −48.4 and −52.3 kJ·mol^−1^/_H2_. The lowest value was found for 2,3-dimethyl-indole (see Table 9, column 5). All other values do not differ significantly from those for unsubstituted indole, and no clear trend can be seen with regard to the position of methylation. Therefore, we can conclude that in terms of $\Delta_{r}H_{m}^{o}$(liq)H_2_-values, all methylated and di-methylated indoles studied in this work can be considered as suitable candidates for LOHC systems.

#### 4.1.2. Energetics of Hydrogen Uptake and Release in an LOHC System Reacting in the Gas Phase from Pure Quantum Chemical Calculations

As a matter of fact, in order to compare the hydrogenation enthalpies, $\Delta_{r}H_{m}^{o}$(liq)H_2_, of methyl- and dimethyl-indoles in the liquid phase, four steps were taken towards these values, as mentioned in the introduction. The time-consuming work was performed to obtain and evaluate reliable enthalpies of vaporization, which combine the *gas-phase* enthalpies of formation, $\Delta_{f}H_{m}^{o}$(gas), and the *liquid-phase* enthalpies of formation, $\Delta_{f}H_{m}^{o}$(liq), of the participants in the hydrogenation reaction. How can the effort for experiments and correlations be reduced? Can some reasonable conclusions about the energetics of hydrogenation/dehydrogenation reactions perhaps already be drawn from the gas-phase enthalpies of formation, $\Delta_{f}H_{m}^{o}$(gas), of the reactants calculated by quantum chemical methods? In other words: how big is the difference between the $\Delta_{r}H_{m}^{o}$(liq)H_2_-values in the *liquid phase* and the $\Delta_{r}H_{m}^{o}$(g)H_2_-values in the *gas phase*? To answer this question, the $\Delta_{r}H_{m}^{o}$(g)H_2_-values for reactions R-I R-II, and R-III were calculated directly from the data given in Table 3 and the results are summarized in Table 10, column 5.

The $\Delta_{r}H_{m}^{o}$(g)H_2_-values were found to be systematically more negative than the corresponding $\Delta_{r}H_{m}^{o}$(liq)H_2_-values, but the difference of about 3 kJ·mol^−1^ is almost constant (see Table 9, column 7), regardless of the position of the methyl substituents at the indole “centerpiece”. Furthermore, the *gas-phase* reaction enthalpies, $\Delta_{r}H_{m}^{o}$(g), of R-I, R-II and R-III (see Table 10) show the same trends compared to the corresponding *liquid-phase* enthalpies,$\;\Delta_{r}H_{m}^{o}$(liq), in Table 9. The differences between the *liquid-phase* and *gas-phase* enthalpies of the reactions of R-I, R-II and R-III (see Appendix A) can be attributed to the differences in the vaporization enthalpies of the HL and HR counterparts of the LOHC systems. However, due to the structural similarity of the reactants on the left and right sides of the hydrogenation reactions, these differences can be roughly assessed and used as a correction to the quantum chemical results, when the energetics of stepwise hydrogenation are of interest.

However, for a general conclusion regarding the amount of hydrogen captured or released by the LOHC system in (kJ·mol^−1^/_H2_)-units, the high-level QC methods provide a clear answer when a series of similarly shaped molecules are of interest.

### 4.2. Thermodynamic Analysis of LOHC Systems Based on Quantum Chemical Methods Only

Not only the energetics of the hydrogenation/dehydrogenation reactions are essential for optimizing hydrogen uptake and release. Since hydrogenation reactions are usually strongly exothermic, they are thermodynamically favorable at room temperature. Due to the exothermic nature of the reaction, the equilibrium can be shifted towards dehydrogenation by increasing the temperature. Therefore, the magnitude of the equilibrium constant can be helpful in locating the temperature range that is suitable for practical applications.

Can we apply modern quantum chemical calculations to determine the order of magnitude of the equilibrium constants at different temperatures and pressures? If so, the understanding of structure–property relationships and the screening of LOHC systems suitable for chemical hydrogen storage will be taken to a higher level. An algorithm leading to the equilibrium constants from QC calculations is presented below. As an example, consider the dehydrogenation of 2-methyl-H8-indole (HR material) to 2-methyl-indole (HL material) with the release of 4 moles of hydrogen, as shown in Figure 9.

For the general reaction “HL” + *n* × H_2_ → “HR”, where HL is the hydrogen-lean compound (or aromatic compound, 2-methyl-indole) and HR is the hydrogen-rich compound (fully hydrogenated aromatic compound, 2-methyl-H8-indole), the equilibrium constant *K* is defined as
*K* = [HR/HL] × [*P*_H2_/*p*_o_]*^n^*
(13)


A pseudo-equilibrium constant, *K*′, is defined as
*K*′ = *K* × [*P*_H2_/*p*_o_]*^n^*
(14)

where *p*_o_ is the standard state pressure 0.1 MPa (i.e., 1 atmosphere) and *K*′ gives the ratio of concentrations of HR to HL. The relationship between the Gibbs energy of reaction and the equilibrium constant is
(15)$$\Delta_{r}G_{m}^{o}(T)=-RT\times \ln\;K$$
where ln *K* denotes the natural logarithm of *K*. Therefore, by rearranging Equation (15),
$$\ln\;K=-\Delta_{r}G_{m}^{o}(T)/RT$$
and from Equation (14),
(16)$$\ln\;K^{\prime }=-\Delta_{r}G_{m}^{o}(T)/RT+[n\times \ln\;(P_{H2}/p_{o})]\;$$

The Gibbs energy of reaction, $\Delta_{r}G_{m}^{o}$, is calculated according to Hess’s law from the Gibbs free energies of formation, $\Delta_{f}G_{m}^{o}$, of the reaction R-IV participants (2-methyl-H8-indole and 2-methyl-indole):(17)$$\Delta_{r}G_{m}^{o}=\Delta_{f}G_{m}^{o}(2\text{-}\mathrm{methyl}\text{-}\mathrm{indole})+\Delta_{f}G_{m}^{o}(H_{2})-\Delta_{f}G_{m}^{o}(2\text{-}\mathrm{methyl}\text{-}H8\text{-}\mathrm{indole})\;$$
with $\Delta_{f}G_{m}^{o}$(H_2_) = 0 kJ·mol^−1^ by definition.

The basic thermodynamic equation for Gibbs energy of formation
(18)$$\Delta_{f}G_{m}^{o}=\Delta_{f}H_{m}^{o}-T\times \Delta_{f}S_{m}^{o}$$
is used to derive $\Delta_{f}G_{m}^{o}$ of individual compounds. The standard molar enthalpies of formation $\Delta_{f}H_{m}^{o}$(g, 298.15 K) of 2-methyl-H8-indole and 2-methyl-indole can be calculated using the high-level QC method, as described in Section 3.1. The standard molar entropies, $S_{m}^{o}$(g, 298.15 K), of 2-methyl-H8-indole and 2-methyl-indole can also be calculated using the QC method, as described in our previous work [3]. The standard molar entropies of formation, $\Delta_{f}S_{m}^{o}$(g, 298.15 K), of 2-methyl-H8-indole and 2-methyl-indole were calculated based on the reaction
*a* C_graphite_ + (*b/*2) H_2_(g) + (*c/*2) N_2_(g) = C_a_H_b_N_c_ (g)
(19)

using the $S_{m}^{o}$-values and the values of entropy of formation for elements: for C_graphite_ (5.74 ± 0.13) J·K^−1^·mol^−1^, for H_2_(g) (130.52 ± 0.02) J·K^−1^·mol^−1^, and for N_2_(g) (191.61 ± 0.01) J·K^−1^·mol^−1^ recommended by Chase [41].

Thus, the Gibbs energy of formation of 2-methyl-H8-indole and 2-methyl-indole can already be calculated at the reference temperature *T* = 298.15 K either from the combination of the experimental and QC results or purely from the QC calculations. However, since a sufficient degree of dehydrogenation is only achieved at higher temperatures, the thermodynamic data are required at practically relevant temperatures. The ideal gas state thermodynamic properties of 2-methyl-H8-indole and 2-methyl-indole, including the standard molar heat capacities and absolute entropies of individual compounds, were calculated in the temperature range from 300 to 600 K using B3LYP hybrid density functional theory with the cc-pvtz(D3BJ) basis set with a “rigid rotator–harmonic oscillator” approach [42,43]. The essential details of the calculation are given in ESI. Results are given in Table 11 and Table 12.

The standard molar enthalpy of formation for the compound from its elements at temperature *T* (taken from [44]) was calculated from Equation (20):(20)$$\Delta_{f}H_{m}^{o}(T)=\Delta_{f}H_{m}^{o}(298\;K)+(H_{T}^{o}-H_{298}^{o}{)}_{\mathrm{compound}}-\Sigma \;(H_{T}^{o}-H_{298}^{o}{)}_{\mathrm{elements}}$$
in which the summation is over the constituent elements in the compound. The standard molar entropy of formation for the compound from its elements at temperature *T* (taken from [44]) was calculated from Equation (21):(21)$$\Delta_{f}S_{m}^{o}(T)=S_{m}^{o}(T{)}_{\mathrm{compound}}-\Sigma \;S_{m}^{o}(T{)}_{\mathrm{elements}}$$

The standard molar Gibbs energy of formation was calculated from the following relation:(22)$$\Delta_{f}G_{m}^{o}(T)=\Delta_{f}H_{m}^{o}(T)-T\times \Delta_{f}S_{m}^{o}(T)$$

The temperature dependencies of the standard molar heat capacities of the gas phase calculated with the QC method were approximated by Equation (23):(23)$$C_{p,m}^{o}(T)=a+b\times T+c\times T^{2}$$
and the approximation coefficients were used to calculate the required thermodynamic functions:(24)$$(H_{T}^{o}-H_{298}^{o})=a\times (T-298)+(b/2)\times (T^{2}-298^{2})+(c/3)\times (T^{3}-298^{3})$$
(25)$$S_{m}^{o}(T)=S_{m}^{o}(298)+a\times \ln\;(T/298)+b\times (T-298)+(c/2)\times (T^{2}-298^{2})$$

The standard molar Gibbs energies of formation, $\Delta_{f}G_{m}^{o}$(*T*), calculated according to Equation (22) for 2-methyl-H8-indole and 2-methyl-indole are given in Table 11 and Table 12 (last columns). These values can now be substituted into Equation (17) to calculate the Gibbs energies of reaction, $\Delta_{f}G_{m}^{o}$(*T*), for the reaction R-IV. The results of these calculations are summarized in Table 13, Table 14 and Table 15.

The resulting Gibbs energies of reaction; $\Delta_{f}G_{m}^{o}$(*T*), at different temperatures are given in these three tables in column 4, so that the ln *K*′-values can now be calculated according to Equation (16). Figure 10 gives the ln *K*′ results from Table 13 (0.1 MPa hydrogen pressure) plotted versus l000/T.

As shown in Figure 10, values of ln *K*′ greater than zero (i.e., *K*′ > 1) denote the thermodynamic conditions favoring 2-methyl-8H-indole formation; values less than zero (i.e., *K*′ < 1) denote reaction conditions favoring 2-methyl-indole formation. An additional advantage of the thermodynamic results derived in Table 13, Table 14 and Table 15 is that they enable the assessment of the influence of pressure on the equilibrium in the LOHC system. Indeed, according to Equation (16), the pressure increase can be screened using the [*n* × ln(*P*_H2_/*p*_o_)]-term. In Table 13, the hydrogen pressure *P*_H2_ = 0.1 MPa was set. In Table 14 and Table 15, this pressure was set to 1 MPa and 2 MPa. Analysis of the *K*′-results in Table 13, Table 14 and Table 15 shows that increasing the hydrogen pressure from 0.1 MPa to 2 MPa allows the dehydrogenation temperature to be reduced from 460 K to 400 K. Thus, the thermodynamic results derived in Table 11, Table 12, Table 13, Table 14 and Table 15 enable the optimization of the experimental conditions of hydrogenation/dehydrogenation of the LOHC systems with help of QC calculations.

The pseudo-equilibrium constants *K*′ calculated in Table 13, Table 14 and Table 15 can be used to assess the thermodynamic equilibrium concentrations of the hydrogen-lean and hydrogen-rich reaction products. In the ideal gas mixture of reaction participants, the partial pressure *P_i_* of each constituent is equal to its mole fraction *N_i_* times the total pressure *P*_tot_:
*P_i_* = *N_i_* × *P*_tot_
(26)


The mole fraction is the number of moles of each constituent, *n_i_*, divided by the total number of moles, *n*_tot_, and the gas-phase constant *K*′ can be expressed as
(27)$$K^{\prime }=K_{n}{\left(\frac{P_{\mathrm{tot}}}{n_{\mathrm{tot}}}\right)}^{\Delta \nu }$$
in which $K_{n}$ is the equilibrium constant expressed in number of moles and Δν is the increment in number of moles of gas in the reaction; that is
$$\Delta \nu =\sum \nu_{i}$$
with $\nu_{i}$ positive for products and negative for reactants. For a typical dehydrogenation of 2-methyl-H8-indole to 2-methyl-indole (reaction R-IV in Figure 9) to release 4 moles of hydrogen. the pseudo-equilibrium constant *K*′ in the standard state is expressed as
(28)$$K^{\prime }=\frac{\left(n_{2Me-indole}\right)\cdot {\left(n_{H2}\right)}^{4}}{n_{2Me-H8-indole}}{\left(\frac{1}{n_{\mathrm{tot}}}\right)}^{4}$$

With the numerical *K*′ values for each reaction temperature from Table 13, Table 14 and Table 15, the composition of the reaction mixture can be estimated by successive approximation.

### 4.3. Empirical Simplification Instead of Quantum Chemical Calculations?

The algorithm and methods for determining the thermodynamic properties of the hydrogen-poor and hydrogen-rich counterparts of the LOHC systems at the reference temperature T = 298.15 K are well established [45]. A reasonable combination of experimental, empirical and theoretical methods leads to reliable results for $\Delta_{f}H_{m}^{o}$(298.15 K), $\Delta_{f}S_{m}^{o}$(298.15 K), $S_{m}^{o}$(298.15 K) and finally for $\Delta_{f}G_{m}^{o}$(298.15 K). The determination of these thermodynamic properties at elevated temperatures is a challenging task since the calculation of the heat capacities of the ideal gases requires not only a large amount of auxiliary information, but also profound experience in such calculations. In our recent work [3], the ideal-gas heat capacities, $C_{p,m}^{o}$, of indole, indoline, H8-indole, 2-methyl-indole, 2-methyl-indoline and 2-methyl-H8-indole were calculated at different temperatures between 300 and 600 K using QC methods. It has turned out that for all these indoles the temperature dependences of the heat capacities $C_{p,m}^{o}=f\left(T\right)$ are not linear in the range 300–600 K (see Appendix A).

However, it can be noticed that the gradients for the aromatic counterparts of the LOHC systems (indole, indoline, 2-methyl-indole and 2-methyl-indoline) were not very different. The same observation was made for the aliphatic counterparts of the LOHC systems (H8-indole, 2-methyl-H8-indole); their gradients were also very similar.

It was possible to obtain the linear temperature dependencies $C_{p,m}^{o}=f\left(T\right)$ when the temperature axis was used as a logarithmic function:(29)$$C_{p,m}^{o}=a\cdot ln\left(T\right)+b$$
where coefficients *a* and *b* are adjustable parameters. The heat capacities of the aromatic counterparts of the LOHC systems (indole, indoline, 2-methyl-indole and 2-methyl-indoline) approximated by Equation (29) are shown in Figure 11.

The heat capacities of the aliphatic counterparts of the LOHC systems (H8-indole, 2-methyl-H8-indole) approximated by Equation (29) are shown in Appendix A. The adjustable parameters *a* and *b* of Equation (29) and the correlation coefficient *R*^2^ for each compound are summarized in Table 16.

As can be seen from Table 16, the coefficients representing the slopes of the temperature dependence of the heat capacity for aromatic materials are in fair agreement. The average value *a* = 169 was taken as a constant parameter for this type of material. For fully hydrogenated indoles (2Me-8H-indole and 8H-indole), the fluctuation of the *a*-values can also be considered acceptable, and the average value *a* = 240 was assumed as a constant parameter for fully hydrogenated indoles. To prove the validity of these assumptions, the heat capacity values for all six indoles were calculated using Equation (29) with the fixed *a*-parameters. These estimates agree well with the original data (see Appendix A).

It can be assumed that the *a*-parameters are also applicable to other types of HR and HL materials (e.g., alkyl-indoles, alkyl-quinolines, alkyl-carbazoles). To apply this simplified method, only $C_{p,m}^{o}$(298.15 K) has to be calculated either with QC methods or with GA methods (e.g., those developed by Benson [39] or by Domalski [46]). With this numerical $C_{p,m}^{o}$(298.15 K)-value, the parameter *b* from Equation (29) is first determined at 298.15 K. Then the $C_{p,m}^{o}=f\left(T\right)$-values for the range of practically relevant temperatures can be estimated. Consequently, these ideal-gas heat capacities can be used in combination with the thermodynamic data at the reference temperature to calculate the *K*′-values, as shown in Section 4.2, which help to optimize the hydrogenation/dehydrogenation reaction conditions.

## 5. Conclusions

A consistent dataset for the thermochemical properties of methyl-substituted indoles has been determined by an approach based on different methods for mutual validation: Experimental measurements utilizing combustion calorimetry and differential scanning calorimetry have been combined with quantum chemical methods and a group-additivity approach. The results confirmed the lower enthalpy of reaction for dehydrogenation in the five-membered ring and the positive effect of lower enthalpy of reaction by the addition of the nitrogen-heteroatom compared to the corresponding homocyclic compound. However, methylation of the indole molecule also has a namable effect on the energetics of the reaction. The overall enthalpy of reaction varies within a range of 18.2 kJ·mol^−1^ (corresponding to 4.6 kJ·mol(H_2_^−1^). The results show that both the degree of methylation and the position of the methyl groups can have a significant influence on the enthalpy of reaction. This not only influences the heat demand for hydrogen release, but also influences the reaction conditions as it influences the equilibrium constant.

## Figures and Tables

**Figure 1 materials-16-02924-f001:**
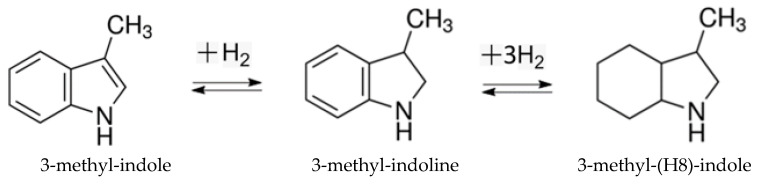
Reversible hydrogenation reactions in the methyl-indole-based LOHC systems studied in this work.

**Figure 2 materials-16-02924-f002:**
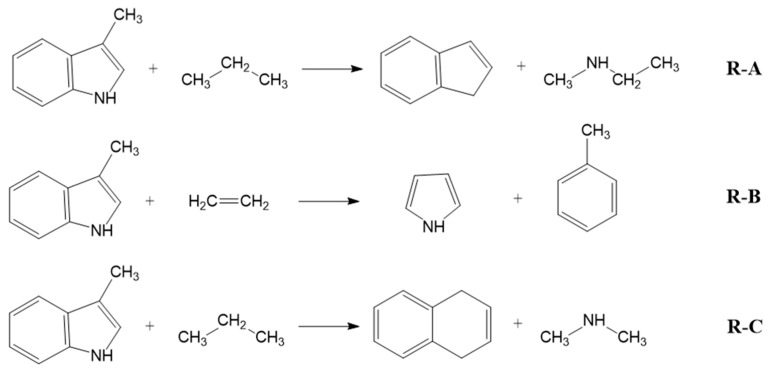
The homodesmotic reactions used to convert the *H*_298_-values to the *theoretical* standard molar enthalpies of formation $\Delta_{f}H_{m}^{o}$(g, 298.15 K).

**Figure 3 materials-16-02924-f003:**
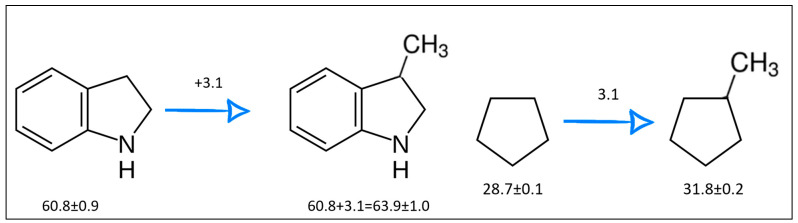
Left: application of the “centerpiece” approach to estimate the vaporization enthalpy of 3-methyl-indoline from the reliable enthalpy of vaporization of indoline [3]. Right: estimation of the CH_3_-contribution from vaporization enthalpies of methyl-cyclopentane [40] and cyclopentane [40]. Numerical values are given in kJ·mol^−1^ at *T* = 298.15 K.

**Figure 4 materials-16-02924-f004:**
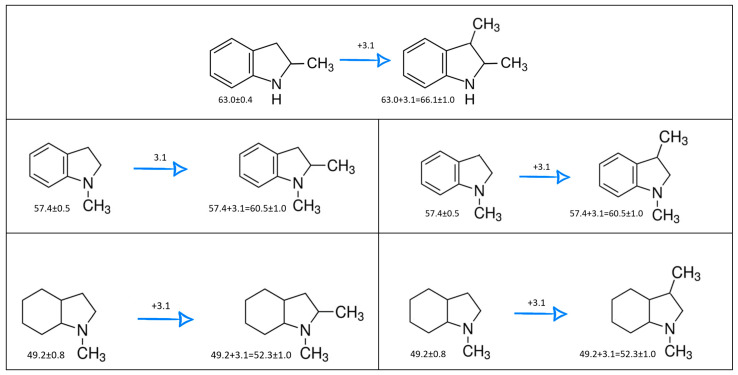
Application of the “centerpiece” approach to estimate the vaporization enthalpies, $\Delta_{l}^{g}H_{m}^{o}$(298.15 K), of 2,3-dimethyl-indoline, 1,2-dimethyl-indoline 1,3-dimethyl-indoline, 1,2-dimethyl-(H8)-indole and 1,3-dimethyl-(H8)-indole. Numerical values are given in kJ·mol^−1^ at *T* = 298.15 K.

**Figure 5 materials-16-02924-f005:**
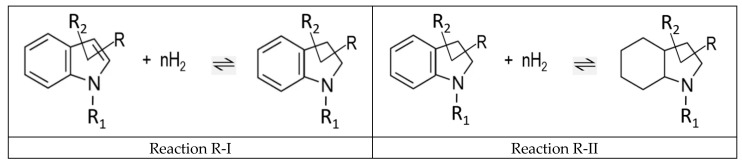
Reversible reactions of the partial hydrogenation (reaction R-I) and the consequent full hydrogenation of the indole derivatives (reaction R-II).

**Figure 6 materials-16-02924-f006:**
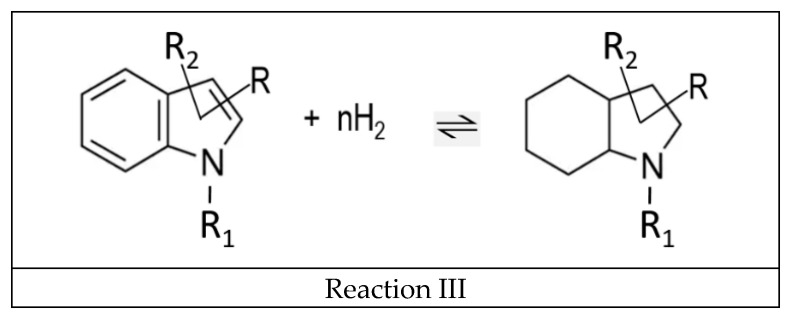
Reversible reactions of the full hydrogenation of the indole derivatives.

**Figure 7 materials-16-02924-f007:**
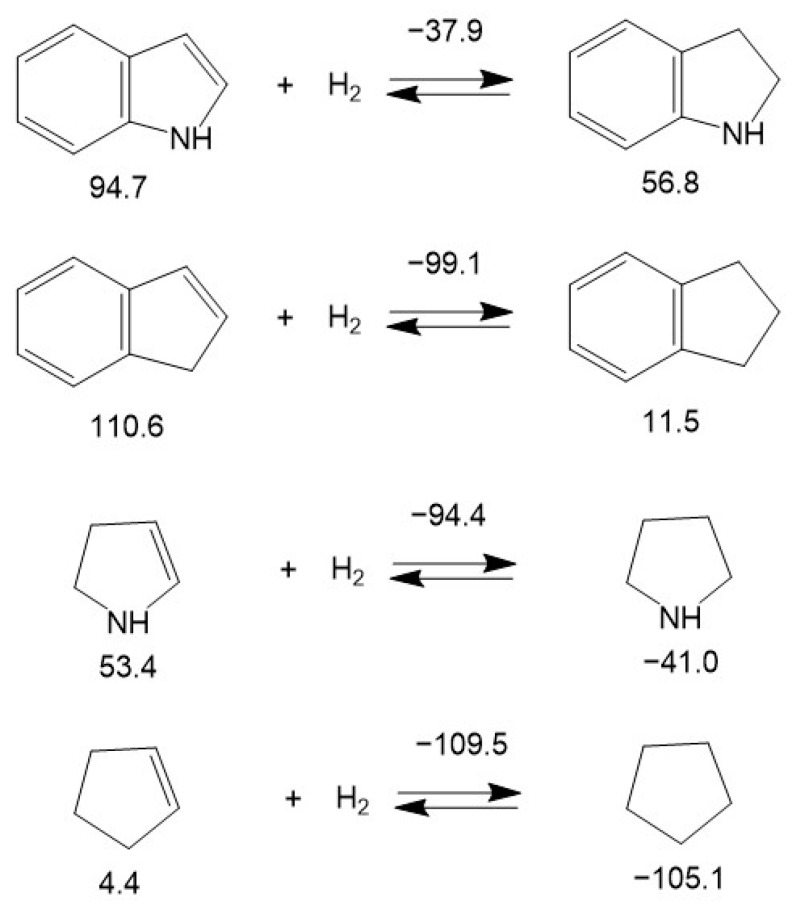
Comparison of the energetics of hydrogenation of double bonds in nitrogen-containing five-membered rings (indole and 2-pyroline) and in the corresponding carbocyclic aromatic compounds (indene and cyclopentene). Figures below molecules are $\Delta_{f}H_{m}^{o}$(liq) taken from Appendix A. Figures above the arrows are $\Delta_{r}H_{m}^{o}$(liq) calculated according to Equation (1). All values are in kJ·mol^−1^.

**Figure 8 materials-16-02924-f008:**
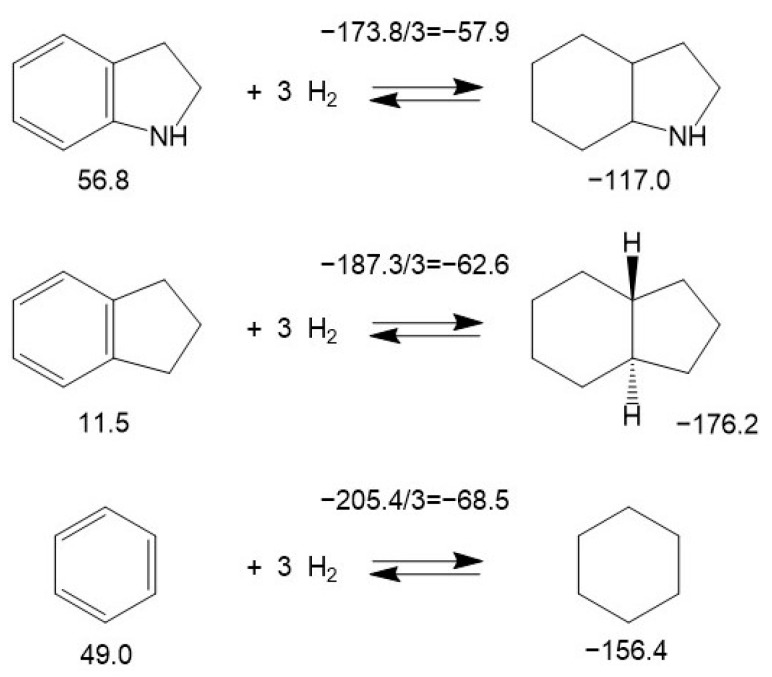
Comparison of the energetics of hydrogenation of three double bonds in the benzene ring of indole, in indane and in benzene. Figures below molecules are $\Delta_{f}H_{m}^{o}$(liq) taken from Appendix A. Figures above the arrows are $\Delta_{r}H_{m}^{o}$(liq) calculated according to Equation (1) and divided by the number of double bonds. All values are in kJ·mol^−1^.

**Figure 9 materials-16-02924-f009:**
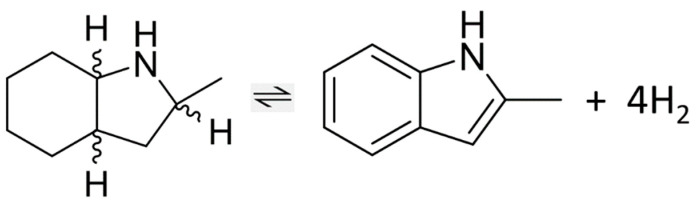
Dehydrogenation of 2-methyl-H8-indole to 2-methyl-indole (reaction R-IV) to release 4 moles of hydrogen.

**Figure 10 materials-16-02924-f010:**
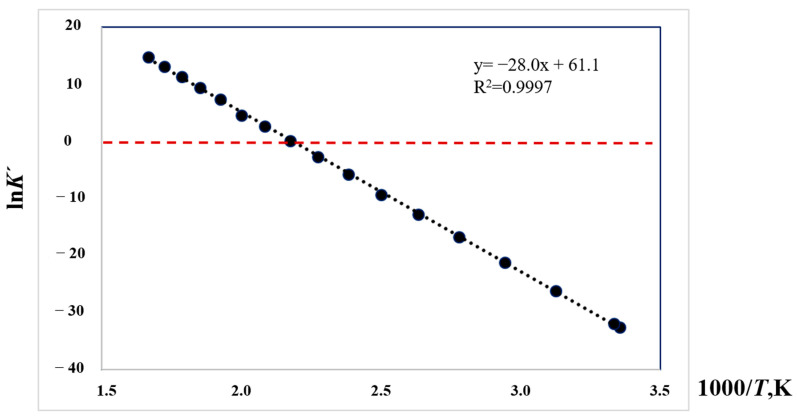
The ln *K*′ values as a function of inverse temperature for the dehydrogenation of 2-methyl-8H-indole to 2-methyl-indole (calculated from the results in Table 13).

**Figure 11 materials-16-02924-f011:**
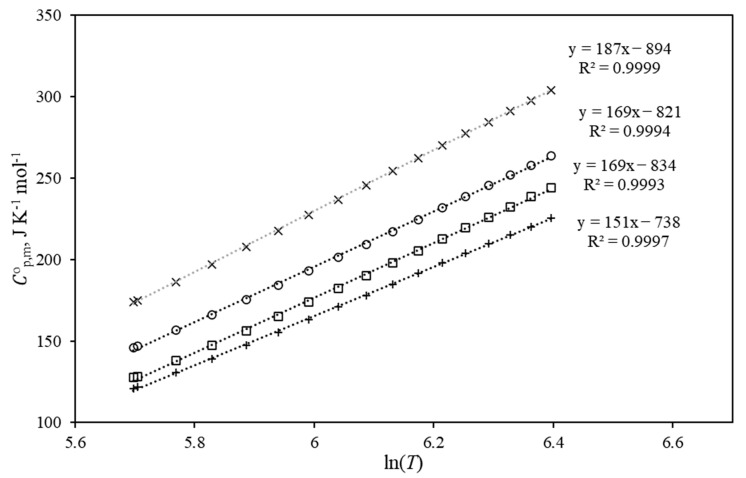
The gas-phase heat capacities of the aromatic counterparts of the LOHC systems approximated by Equation (29): □—indoline, ○—2Me-indole, ×—2-Me-indoline, +—indole.

**Table 1 materials-16-02924-t001:** Compilation of thermochemical data for indole derivatives at *T* = 298.15 K (*p* = 0.1 MPa, in kJ·mol^−1^) ^a^.

Compound	$\Delta_{f}H_{m}^{o}$(cr,l)	$\Delta_{cr,l}^{g}H_{m}^{o}$	$\Delta_{f}H_{m}^{o}$(g)_exp_	$\Delta_{f}H_{m}^{o}$(g)_G4_ ^b^
1	2	3	4	5
indole (cr) [3]	87.2 ± 0.9	75.3 ± 0.4	162.5 ± 1.0	160.4
indoline (liq) [3]	60.0 ± 0.9	60.8 ± 0.9	120.8 ± 1.3	117.6
H8-indole (liq) [3]	−117.5 ± 1.8	53.5 ± 0.7	−64.0 ± 1.9	−63.5
2-methyl-indole (cr) [3]	36.1 ± 1.3	85.3 ± 0.4	121.4 ± 1.4	120.2
2-methyl-indoline [3]	17.2 ± 1.9	63.0 ± 0.4	80.2 ± 1.9	79.4
2-methyl-H8-indole [3]	−157.1 ± 2.1	57.8 ± 0.8	−99.3 ± 2.2	−100.4
3-methyl-indole (cr)	47.4 ± 2.3 [24]	81.3 ± 0.5 ^c^	128.7 ± 2.4	126.0 ^d^
2,3-dimethyl-indole (cr)	4.2 ± 1.0 [25]	86.0 ± 0.6 ^c^	90.2 ± 1.2	86.6 ^d^

^a^ The uncertainties in this table are given as 2 times the standard deviation. ^b^ Calculated in [3] with the G4 method using the atomization procedure. ^c^ Evaluated in this work. ^d^ Calculated in this work with the G4 method using the atomization procedure.

**Table 2 materials-16-02924-t002:** Results of the quantum chemical calculations of $\Delta_{f}H_{m}^{o}$(g, 298.15 K) for 3-methyl-indole with different methods at *T* = 298.15 K (*p°* = 0.1 MPa, in kJ·mol^−1^).

Method	(AT)	R-A	R-B	R-C	Exp. [Table 1]
G4 ^a^	126.0	126.6	125.6	126.6	128.7 ± 2.4
G3MP2 ^b^	122.9	129.1	127.2	129.3	
CBS-APNO ^c^	125.1	127.8	125.4	127.5	

^a^ Calculated according to the G4 method using atomization reaction Equation (4) and the homodesmotic reactions shown in Figure 2. The expanded uncertainties are assessed to be ±3.5 kJ·mol^−1^ [18]. ^b^ Calculated according to the G3MP2 method using atomization reaction Equation (4) and the homodesmotic reactions shown in Figure 2. The expanded uncertainties are assessed to be ±4.1 kJ·mol^−1^ [17]. ^c^ Calculated according to the CBS-APNO method using atomization reaction Equation (4) and the homodesmotic reactions shown in Figure 2. The expanded uncertainties are assessed to be ±4.1 kJ·mol^−1^ [19].

**Table 3 materials-16-02924-t003:** Structures of the most stable conformers and the G4 calculated gas-phase enthalpies of formation $\Delta_{f}H_{m}^{o}$(g)_G4_ at *T* = 298.15 K (*p°* = 0.1 MPa) for indole derivatives (in kJ·mol^−1^).

Indole	Structures	$\Delta_{f}H_{m}^{o}$(g)_G4_ ^a^
3-methyl-indole 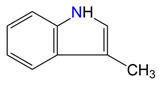	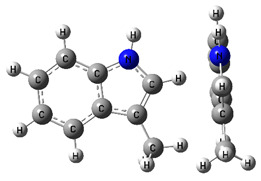	126.0
3-methyl-indoline 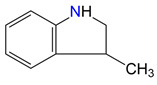	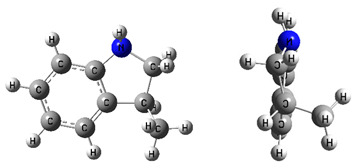	85.8
cis-3-methyl-(H8)-indole 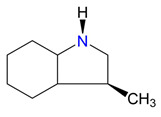	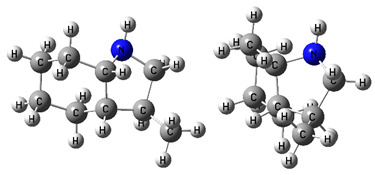	−93.7
trans-3-methyl-(H8)-indole 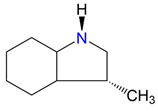	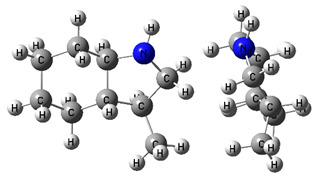	−93.8
1,2-dimethylindole * 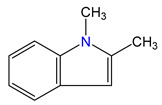 *	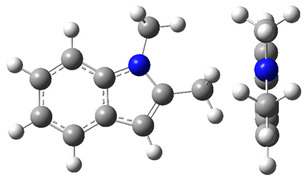	111.4
1,2-dimethyl-indoline 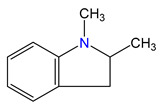	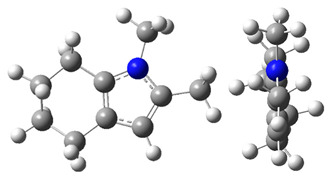	72.8
cis-1,2-dimethyl-(H8)-indole 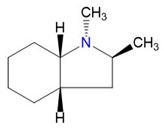	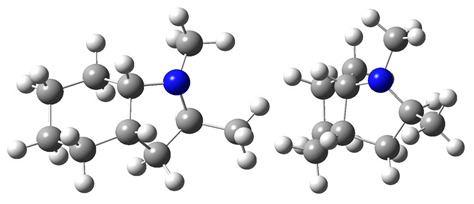	−105.5
trans-1,2-dimethyl-(H8)-indole 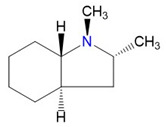	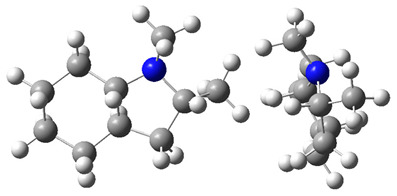	−113.0
1,3-dimethylindole 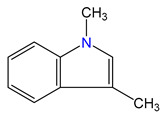	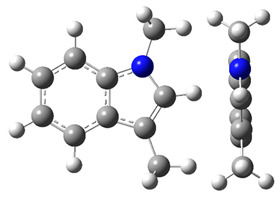	116.3
1,3-dimethyl-indoline 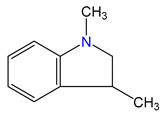	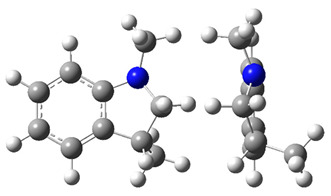	74.6
1,3-dimethyl-(H8)-indole 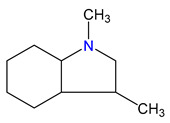	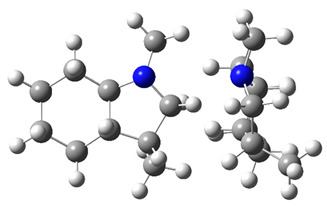	−102.4
2,3-dimethylindole 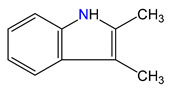	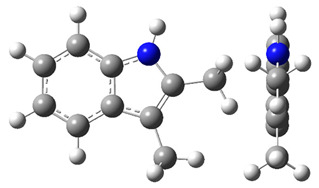	86.6
2,3-dimethyl-indoline 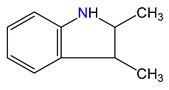	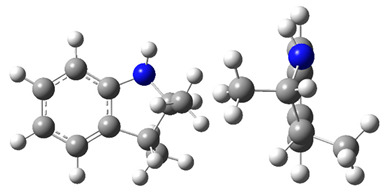	47.0
2,3-dimethyl-(H8)-indole 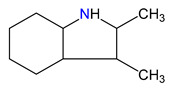	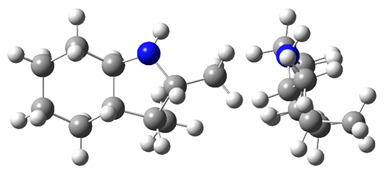	−120.9

^a^ Calculated according to the G4 method using the atomization procedure.

**Table 4 materials-16-02924-t004:** Results of transpiration method for methyl-indole derivatives: absolute vapor pressures *p*, standard molar sublimation/vaporization enthalpies and standard molar sublimation/vaporization entropies.

*T*/ K ^a^	*m*/ mg ^b^	*V*(N_2_) ^c^/ dm^3^	*T*_a_/ K ^d^	Flow/ dm^3^·h^−1^	*p*/ Pa ^e^	*u*(*p*)/ Pa ^f^	$\Delta_{cr,l}^{g}H_{m}^{o}$(*T*)/ kJ·mol^−1^	$\Delta_{l,cr}^{g}S_{m}^{o}$(*T*)/ J·K^−1^·mol^−1^
3-methyl-indole: $\Delta_{\mathrm{cr}}^{g}H_{m}^{o}$(298.15 K) = (81.4 ± 0.6) kJ·mol^−1^								
$\ln\;\left(p/p_{ref}\right)=\frac{296.4}{R}-\frac{89468.6}{RT}-\frac{27.1}{R}\ln\frac{T}{298.15}$; *p*_ref_ = 1 Pa								
285.2	1.08	143.6	296.2	4.03	0.14	0.01	81.7	174.7
288.2	1.13	104.3	296.2	4.03	0.20	0.01	81.7	174.4
290.2	1.31	93.58	296.2	4.03	0.26	0.01	81.6	174.4
293.2	1.26	64.12	298.2	4.05	0.37	0.01	81.5	174.2
298.2	1.53	44.26	298.2	2.95	0.65	0.02	81.4	173.7
308.2	1.60	16.20	296.2	4.05	1.85	0.05	81.1	172.6
315.8	1.29	6.038	296.2	4.03	4.01	0.11	80.9	172.0
1,2-dimethyl-indole: $\Delta_{\mathrm{cr}}^{g}H_{m}^{o}$(298.15 K) = (83.2 ± 0.7) kJ·mol^−1^								
$\ln\;\left(p/p_{ref}\right)=\frac{307.8}{R}-\frac{92463.2}{RT}-\frac{31.1}{R}\ln\frac{T}{298.15}$; *p*_ref_ = 1 Pa								
298.2	2.12	47.790	293.2	3.19	0.75	0.02	83.2	180.9
302.2	0.84	11.90	293.2	5.10	1.19	0.03	83.1	180.6
306.2	0.83	7.653	293.2	5.10	1.81	0.05	82.9	180.1
310.2	0.85	5.102	293.2	5.10	2.81	0.08	82.8	179.9
312.2	0.76	3.797	293.2	5.06	3.36	0.09	82.8	179.5
314.2	0.34	1.367	293.2	4.10	4.13	0.11	82.7	179.3
314.2	0.72	2.953	293.2	5.06	4.09	0.11	82.7	179.2
316.2	0.81	2.700	293.2	5.06	5.06	0.15	82.6	179.1
318.2	0.62	1.709	293.2	4.10	6.13	0.18	82.6	178.9
320.2	0.73	1.645	293.2	5.06	7.48	0.21	82.5	178.7
322.2	0.75	1.367	293.2	4.10	9.18	0.25	82.4	178.6
324.2	0.94	1.435	293.2	5.06	10.94	0.30	82.4	178.3
326.2	1.08	1.367	293.2	4.10	13.25	0.36	82.3	178.2
1,2-dimethyl-indole: $\Delta_{l}^{g}H_{m}^{o}$(298.15 K) = (67.6 ± 0.7) kJ·mol^−1^								
$\ln\;\left(p/p_{ref}\right)=\frac{300.8}{R}-\frac{88718.3}{RT}-\frac{70.8}{R}\ln\frac{T}{298.15}$; *p*_ref_ = 1 Pa								
331.2	1.98	1.543	293.2	2.06	21.59	0.56	65.3	126.9
334.2	1.91	1.200	293.2	2.06	26.66	0.69	65.1	126.3
338.2	1.76	0.857	293.2	2.06	34.43	0.89	64.8	125.3
342.2	1.85	0.686	293.2	2.06	45.34	1.16	64.5	124.5
346.2	2.45	0.698	293.2	2.09	59.00	1.50	64.2	123.7
350.2	2.02	0.441	293.2	1.06	76.91	1.95	63.9	123.0
354.2	2.07	0.353	293.2	1.06	98.12	2.48	63.6	122.1
358.2	2.51	0.344	293.2	1.06	122.29	3.08	63.4	121.2
362.2	2.63	0.282	293.2	1.06	155.94	3.92	63.1	120.4
366.2	5.08	0.423	293.2	1.01	201.38	5.06	62.8	119.9
370.2	3.75	0.254	293.2	1.01	247.93	6.22	62.5	119.0

^a^ Saturation temperature measured with the standard uncertainty (*u(T*) = 0.1 K). ^b^ Mass of transferred sample condensed at T = 243 K. ^c^ Volume of nitrogen (*u*(*V*) = 0.005 dm^3^) used to transfer *m* (*u*(*m*) = 0.0001 g) of the sample. Uncertainties are given as standard uncertainties. ^d^ Ta is the temperature of the soap bubble meter used for measurement of the gas flow. ^e^ Vapor pressure at temperature *T*, calculated from the *m* and the residual vapor pressure at the condensation temperature calculated by an iteration procedure. ^f^ Standard uncertainties were calculated with *u*(*p_i_*/Pa) = 0.005 + 0.025 (*p_i_*/Pa) for pressures below 5 Pa and with *u*(*p_i_*/Pa) = 0.025 + 0.025 (pi/Pa) for pressures from 5 to 3000 Pa. The standard uncertainties for *T*, *V*, *p* and *m* correspond to a confidence level of 68.3%. Uncertainties of the sublimation/vaporization enthalpies *U*($\Delta_{\mathrm{cr},l}^{g}H_{m}^{o}$) are the expanded uncertainty (level of confidence: 95%, corresponding to 2 times the standard deviation) calculated according to procedures described elsewhere [28]. Uncertainties include uncertainties from the experimental conditions and the fitting equation, vapor pressures and uncertainties from adjustment of vaporization enthalpies to the reference temperature *T* = 298.15 K.

**Table 5 materials-16-02924-t005:** Compilation of available enthalpies of sublimation/vaporization $\Delta_{\mathrm{cr},l}^{g}H_{m}^{o}$ of methyl-substituted indole derivatives.

Compound/CAS	Method ^a^	*T*-Range	$\Delta_{cr,l}^{g}H_{m}^{o}$(*T*_av_)	$\Delta_{cr,l}^{g}H_{m}^{o}$(298.15 K) ^b^	Ref.
		K	kJ·mol^−1^	kJ·mol^−1^	
3-methyl-indole (cr II)	n/a	288–333	83.3 ± 2.0	(83.6 ± 2.1) ^c^	[29]
83-34-1	DC	349.9	83.0 ± 1.9	(90.4 ± 2.4)	[24]
	T	285.2–315.8	81.4 ± 0.6	81.4 ± 0.6	Table 4
	FT			81.2 ± 0.7	Appendix A
				**81.3 ± 0.5** ^d^	average
3-methyl-indole (cr I)	T	317.2–364.2	78.0 ± 0.4	79.3 ± 0.5	[3]
3-methyl-indole (liq)	n/a	368.2–539.4	63.7 ± 1.0	73.8 ± 2.2	[30]
	FT			69.8 ± 0.6	Appendix A
	FT			70.0 ± 1.3	Appendix A
	*J_x_*			70.2 ± 1.5	[3]
	*J_Lee_*			70.2 ± 1.5	[3]
				**70.1 ± 0.5** ^d^	average
2,3-dimethyl-indole (cr)				**86.0 ± 0.6**	[3]
91-55-4					
2,3-dimethyl-indole (liq)				**75.2 ± 1.0**	[3]
1,2-dimethyl-indole (cr)	T	298.2–326.2	82.8 ± 0.6	83.2 ± 0.7	Table 4
1,2-dimethyl-indole (liq)	T	331.2–370.2	64.0 ± 0.6	67.6 ± 0.7	Table 4
875-79-6	FT			69.9 ± 1.3	Appendix A
	*J_Lee_*			67.6 ± 1.0	Appendix A
				**67.6 ± 0.6** ^d^	average
1,3-dimethyl-indole (liq)	BP	335–533	58.7 ± 1.2	67.9 ± 1.2	Appendix A
875-30-9	*T* _b_			67.3 ± 1.0	Table 4
				**67.5 ± 0.8** ^d^	average
3-methyl-indoline (liq)	BP	329–525	56.6 ± 0.9	65.1 ± 1.9	Appendix A
4375-15-9	*J_x_*			63.2 ± 1.0	Table 6
	*J_x_*			64.1 ± 1.5	[3]
	CP			63.9 ± 1.0	this work
				**63.8 ± 0.6** ^d^	average
2,3-dimethyl-indoline (liq)	BP	398–523	54.9 ± 3.0	67.1 ± 3.9	Appendix A
22120-50-9	CP			66.1 ± 1.0	this work
				**66.2 ± 1.0** ^d^	average
1,2-dimethyl-indoline (liq)	BP	314–501	53.1 ± 1.8	60.8 ± 2.3	Appendix A
26216-93-3	CP			60.5 ± 1.0	this work
				**60.5 ± 0.9** ^d^	average
1,3-dimethyl-indoline (liq)	BP	322–523	51.5 ± 1.6	60.0 ± 2.3	Appendix A
39891-78-6	CP			60.5 ± 1.0	this work
				**60.4 ± 0.9** ^d^	average
3-methyl-(H8)-indole (liq)	*J_x_*			58.0 ± 1.0	Table 6
85158-21-0	*J_x_*			58.1 ± 1.0	[3]
				**58.0 ± 0.7** ^d^	average
2,3-dimethyl-(H8)-indole (liq)	*J_x_*			61.2 ± 1.0	[3]
1394248-06-6					
1,2-dimethyl-(H8)-indole (liq)	CP			52.3 ± 1.0	this work
87401-40-9					
1,3-dimethyl-(H8)-indole (liq)	CP			52.3 ± 1.0	this work
87401-41-0					

^a^ Techniques: DC = drop calorimetry; T = transpiration method; FT = derived as the difference of sublimation and fusion enthalpies (see Appendix A); *J_x_*—derived from correlation with Kovats indices BP = derived from boiling points at different temperatures available in the literature (see Appendix A); *J_Lee_*—derived from correlation with Lee indices (see Appendix A); n/a—method is not available; CP = derived using the “centerpiece” approach (see text); *T*_b_ = derived from correlation of vaporization enthalpies with the normal boiling points (see text). ^b^ Uncertainty of the sublimation/vaporization enthalpy *U*($\Delta_{\mathrm{cr},l}^{g}H_{m}^{o}$) is the expanded uncertainty (0.95 level of confidence, k = 2) calculated according to a procedure described elsewhere [28]. It includes uncertainties from the experimental conditions, uncertainties of vapor pressure, uncertainties from the fitting equation and uncertainties from temperature adjustment to *T* = 298.15 K. ^c^ The phase transition at *T*_tr_ = 316.8 K and enthalpy $\Delta_{\mathrm{crII}}^{\mathrm{crI}}H_{m}^{o}$ = 2.7 ± 0.6 kJ·mol^−1^ was reported in our previous work [3]. ^d^ Weighted mean value. Values in parenthesis were excluded from the calculation of the mean. Values in bold are recommended for further thermochemical calculations.

**Table 6 materials-16-02924-t006:** Correlation of the vaporization enthalpies, $\Delta_{l}^{g}H_{m}^{o}$(298.15 K), of cyclic aliphatic and aromatic hydrocarbons with their Kovats indices (*J_x_*).

Compound	*J_x_* ^a^	$\Delta_{l}^{g}H_{m}^{o}$(298 K)_exp_	$\Delta_{l}^{g}H_{m}^{o}$(298 K)_calc_ ^b^	Δ ^c^
		kJ·mol^−1^	kJ·mol^−1^	kJ·mol^−1^
1-methyl-pyrrolidine	697	34.2 ± 0.2 [35]	34.0	0.2
toluene	780	38.1 ± 0.2 [35]	37.8	0.3
1,4-dimethylbenzene	876	42.4 ± 0.2 [35]	42.3	0.1
indene	1059	50.6 ± 1.5 [36]	50.7	−0.1
indane	1033	49.2 ± 1.0 [36]	49.5	−0.3
tetraline	1164	55.2 ± 1.0 [35]	55.6	−0.4
quinoline	1231	59.3 ± 0.4 [4]	58.7	0.6
1-methyl-indole	1285	61.9 ± 0.3 [37]	61.2	0.7
H8-indole	1140	53.5 ± 0.7 [3]	54.5	−1.0
3-methyl-indoline	1330		**63.2 ± 1.0**	
3-methyl-(H8)-indole	1217		**58.0 ± 1.0**	

^a^ Kovats indices at 443 K, *J_x_*, on the standard non-polar column SE-30. ^b^ Calculated using Equation (10) with the assessed expanded uncertainty of ±1.0 kJ·mol^−1^ (0.95 level of confidence, k = 2). ^c^ Difference between columns 3 and 4 in this table.

**Table 7 materials-16-02924-t007:** Correlation of the vaporization enthalpies $\Delta_{l}^{g}H_{m}^{o}$(298.15 K) of indole derivatives and their *T_b_* normal boiling temperatures.

	*T_b_* ^a^	$\Delta_{l}^{g}H_{m}^{o}$(298.15 K)_exp_	$\Delta_{l}^{g}H_{m}^{o}$(298.15 K)_calc_ ^b^	Δ ^c^
Compound	K	kJ·mol^−1^	kJ·mol^−1^	kJ·mol^−1^
indole	527.0	65.6 ± 0.4 [4]	65.9	−0.3
1-methyl-indole	512.6	61.9 ± 0.3 [37]	61.6	0.3
2-methyl-indole	545.2	71.7 ± 0.4 [3]	71.4	0.3
3-methyl-indole	538.7	70.1 ± 0.5 [Table 5]	69.4	0.7
5-methyl-indole	540.2	70.4 ± 0.7 [4]	69.9	0.5
7-methyl-indole	539.2	68.8 ± 0.8 [4]	69.6	−0.8
1,2-dimethyl-indole	533.5	67.6 ± 0.7 [Table 4]	67.9	−0.3
2,3-dimethyl-indole	558.2	75.2 ± 1.0 [Table 5]	75.3	−0.1
1,3-dimethyl-indole	531.7		**67.3 ± 1.0**	

^a^ Normal boiling temperatures are from [33]. ^b^ Calculated using Equation (12). ^c^ Difference between columns 3 and 4.

**Table 8 materials-16-02924-t008:** Calculation of the *liquid-phase* enthalpies of formation, $\Delta_{f}H_{m}^{o}$(liq), of the indole derivatives, at *T* = 298.15 K (*p°* = 0.1 MPa, in kJ·mol^−1^).

Compound	$\Delta_{f}H_{m}^{o}$(Gas) ^a^	$\Delta_{l}^{g}H_{m}^{o}$ ^b^	$\Delta_{f}H_{m}^{o}$(liq) ^c^
indole [3]	160.4	65.7 ± 0.6	94.7
indoline [3]	117.6	60.8 ± 0.9	56.8
H8-indole [3]	−63.5	53.5 ± 0.7	−117.0
2-methyl-indole [3]	120.2	71.7 ± 0.4	48.5
2-methyl-indoline [3]	79.4	63.0 ± 0.4	16.4
*trans*-2-methyl-(H8)-indole [3]	−100.4	57.8 ± 0.8	−158.2
3-methyl-indole	126.0	70.1 ± 0.5	55.9
3-methyl-indoline	85.8	63.8 ± 0.6	22.0
*trans*-3-methyl-(H8)-indole	−93.8	58.0 ± 0.7	−151.8
2,3-dimethylindole	86.6	75.2 ± 1.0	11.4
2,3-dimethyl-indoline	47.0	66.2 ± 1.0	−19.2
2,3-dimethyl-(H8)-indole	−120.9	61.2 ± 1.0	−182.1
1,2-dimethylindole	111.4	67.6 ± 0.5	43.8
1,2-dimethyl-indoline	72.8	60.5 ± 0.9	12.3
*trans*-1,2-dimethyl-(H8)-indole	−113.0	52.3 ± 1.0	−165.3
1,3-dimethylindole	116.3	67.5 ± 0.8	48.8
1,3-dimethyl-indoline	74.6	60.4 ± 0.9	14.2
1,3-dimethyl-(H8)-indole	−102.4	52.3 ± 1.0	−154.7

^a^ The G4 calculated values from Table 3. ^b^ Evaluated values from Table 5. ^c^ Calculated according to Equation (2).

**Table 9 materials-16-02924-t009:** Calculation of the *liquid-phase* reaction enthalpies, $\Delta_{r}H_{m}^{o}$(liq), of the hydrogenation of indole derivatives, at *T* = 298.15 K (*p°* = 0.1 MPa, in kJ·mol^−1^).

Substituents	R-I	R-II	R-III			
	$\Delta_{r}H_{m}^{o}$(liq) ^a^	$\Delta_{r}H_{m}^{o}$(liq) ^b^	$\Delta_{r}H_{m}^{o}$(liq) ^c^	$\Delta_{r}H_{m}^{o}$(liq)H_2_ ^d^	$\Delta_{r}H_{m}^{o}$(g)/H_2_ ^e^	Δ ^f^
R = R_1_ = R_2_ = H	−37.9	−173.8	−211.7	−52.9	−56.0	3.1
R = 2-CH_3_; R_1_ = R_2_ = H	−32.1	−174.6	−206.7	−51.7	−55.2	3.5
R = 3-CH_3_; R_1_ = R_2_ = H	−33.9	−173.8	−207.7	−51.9	−55.0	3.1
R = 2-CH_3_; R_1_ = H; R_2_ = 3-CH_3_	−30.6	−162.9	−193.5	−48.4	−51.9	3.5
R = 2-CH_3_; R_1_ = CH_3_; R_2_ = H	−31.7	−177.4	−209.1	−52.3	−56.1	3.8
R = H; R_1_ = CH_3_; R_2_ = CH_3_	−34.8	−168.7	−203.5	−50.8	−54.7	3.9

^a^ Calculated according to Hess’s law applied to R-I using the enthalpies of formation of the reactants from Table 8. ^b^ Calculated according to Hess’s law applied to R-II using the enthalpies of formation of the reactants from Table 8. ^c^ Calculated according to Hess’s law applied to R-III using the enthalpies of formation of the reactants from Table 8. ^d^ Liquid-phase reaction enthalpy per mole H_2_ calculated from data for reaction R-III in this table. ^e^ Gas-phase reaction enthalpy per mole H_2_ calculated from data for reaction R-III in Table 10. ^f^ Difference between columns 5 and 6 in this table.

**Table 10 materials-16-02924-t010:** Calculation of the *gas-phase* reaction enthalpies, $\Delta_{r}H_{m}^{o}$(g), of the hydrogenation of indole derivatives, at *T* = 298.15 K (*p°* = 0.1 MPa, in kJ·mol^−1^).

Substituents	R-I	R-II	R-III	
	$\Delta_{r}H_{m}^{o}$(g) ^a^	$\Delta_{r}H_{m}^{o}$(g) ^b^	$\Delta_{r}H_{m}^{o}$(g) ^c^	$\Delta_{r}H_{m}^{o}$(g)/H_2_ ^d^
R = R_1_ = R_2_ = H	−42.8	−181.1	−223.9	−56.0
R = 2-CH_3_; R_1_ = R_2_ = H	−40.8	−179.8	−220.6	−55.2
R = 3-CH_3_; R_1_ = R_2_ = H	−40.2	−179.6	−219.8	−55.0
R = 2-CH_3_; R_1_ = H; R_2_ = 3-CH_3_	−39.6	−167.9	−207.5	−51.9
R = 2-CH_3_; R_1_ = CH_3_; R_2_ = H	−38.6	−185.8	−224.4	−56.1
R = H; R_1_ = CH_3_; R_2_ = CH_3_	−41.7	−177.0	−218.7	−54.7

^a^ Calculated according to Hess’s law applied to R-I using the enthalpies of formation of the reactants from Table 8. ^b^ Calculated according to Hess’s law applied to R-II using the enthalpies of formation of the reactants from Table 8. ^c^ Calculated according to Hess’s law applied to R-III using the enthalpies of formation of the reactants from Table 8. ^d^ Reaction enthalpy per mole H_2_ calculated from data for reaction R-III.

**Table 11 materials-16-02924-t011:** Gas-phase standard molar thermodynamic properties of 2-methyl-indole at *p* = 0.1 MPa.

*T*	$C_{p,m}^{o}(T)$	$(H_{T}^{o}$ − $H_{298}^{o}{)}_{c}$	$(H_{T}^{o}$ − $H_{298}^{o}{)}_{e}$	$S_{m}^{o}(T{)}_{c}$	$S_{m}^{o}(T{)}_{e}$	$\Delta_{f}H_{m}^{o}(T)$	$\Delta_{f}S_{m}^{o}(T)$	$\Delta_{f}G_{m}^{o}(T)$
K	J·K^−1^·mol^−1^	kJ·mol^−1^	kJ·mol^−1^	J·K^−1^·mol^−1^	J·K^−1^·mol^−1^	kJ·mol^−1^	J·K^−1^·mol^−1^	kJ·mol^−1^
298.15	146.1	0.0	0.0	368.8	734.7	121.6	−365.9	230.7
300	147.0	0.3	0.4	369.8	736.0	121.2	−366.2	231.0
320	156.5	3.2	5.7	379.5	751.7	115.9	−372.2	235.1
340	165.9	6.3	10.0	389.3	765.7	111.7	−376.4	239.6
360	175.1	9.7	14.4	399.1	779.4	107.2	−380.3	244.1
380	184.1	13.2	19.0	408.8	792.8	102.7	−384.0	248.6
400	192.8	16.9	24.2	418.4	804.5	97.4	−386.1	251.8
420	201.3	20.7	28.6	428.1	818.8	93.1	−390.7	257.1
440	209.4	24.7	33.6	437.6	831.4	88.0	−393.8	261.3
460	217.2	28.9	38.7	447.1	843.7	82.9	−396.6	265.3
480	224.7	33.3	44.1	456.5	855.8	77.6	−399.3	269.2
500	231.9	37.8	50.8	465.8	863.9	70.8	−398.1	269.9
520	238.8	42.4	55.1	475.0	879.1	66.5	−404.1	276.6
540	245.4	47.2	60.9	484.2	890.3	60.8	−406.1	280.1
560	251.8	52.1	66.8	493.2	901.3	54.9	−408.1	283.4
580	257.8	57.2	72.8	502.2	912.1	48.8	−409.9	286.5
600	263.7	62.4	79.8	511.0	924.0	41.9	−413.0	289.7

**Table 12 materials-16-02924-t012:** Gas-phase standard molar thermodynamic properties of 2-methyl-H8-indole at *p* = 0.1 MPa.

*T*	$C_{p,m}^{o}(T)$	$(H_{T}^{o}$ − $H_{298}^{o}{)}_{c}$	$(H_{T}^{o}$ − $H_{298}^{o}{)}_{e}$	$S_{m}^{o}(T{)}_{c}$	$S_{m}^{o}(T{)}_{e}$	$\Delta_{f}H_{m}^{o}(T)$	$\Delta_{f}S_{m}^{o}(T)$	$\Delta_{f}G_{m}^{o}(T)$
K	J·K^−1^·mol^−1^	kJ·mol^−1^	kJ·mol^−1^	J·K^−1^·mol^−1^	J·K^−1^·mol^−1^	kJ·mol^−1^	J·K^−1^·mol^−1^	kJ·mol^−1^
298.15	178.6	0.0	0.0	368.8	1256.7	−99.3	−835.7	149.9
300	179.8	0.3	0.6	369.8	1259.0	−99.6	−836.8	151.4
320	193.0	4.1	8.1	379.5	1283.4	−103.4	−849.2	168.4
340	206.1	8.0	14.7	389.3	1304.2	−106.0	−857.9	185.7
360	219.2	12.3	21.5	399.1	1324.6	−108.5	−866.2	203.4
380	232.1	16.8	28.4	408.8	1344.5	−110.9	−873.9	221.2
400	244.8	21.6	36.1	418.4	1361.0	−113.8	−878.1	237.5
420	257.3	26.6	42.6	428.1	1382.9	−115.3	−887.8	257.5
440	269.3	31.9	50.0	437.6	1401.4	−117.4	−894.0	275.9
460	281	37.4	57.5	447.1	1419.4	−119.4	−899.8	294.5
480	292.4	43.1	65.1	456.5	1437.0	−121.3	−905.2	313.2
500	303.3	49.0	74.4	465.8	1446.5	−124.6	−902.5	326.6
520	313.9	55.2	80.9	475.0	1470.8	−125.0	−914.7	350.7
540	324.1	61.5	89.0	484.2	1487.0	−126.7	−918.9	369.5
560	333.9	68.1	97.2	493.2	1502.7	−128.4	−922.6	388.3
580	343.3	74.9	105.6	502.2	1518.0	−130.0	−926.1	407.1
600	352.4	81.8	115.1	511.0	1534.3	−132.5	−930.6	425.8

**Table 13 materials-16-02924-t013:** Gas-phase standard molar Gibbs energies of 2-methyl-8H-indole and 2-methyl-indole and thermodynamic parameters of the dehydrogenation reaction R-IV at *p°* = 0.1 MPa ^a^.

*T*	$\Delta_{f}G_{m}^{o}(T)$ _2-Me-8H-Ind_	$\Delta_{f}G_{m}^{o}(T)$ _2-Me-Ind_	$\Delta_{r}G_{m}^{o}(T)$ _R-IV_	ln *K*′	*K*′
K	kJ·mol^−1^	kJ·mol^−1^	kJ·mol^−1^		
298.15	149.9	230.7	80.8	−32.6	6.9 × 10^−15^
300	151.4	231.3	79.9	−32.0	1.2 × 10^−14^
320	168.4	238.3	69.9	−26.3	3.9 × 10^−12^
340	185.7	246.0	60.2	−21.3	5.6 × 10^−10^
360	203.4	253.8	50.4	−16.8	4.8 × 10^−8^
380	221.2	261.7	40.5	−12.8	2.7 × 10^−6^
400	237.5	268.7	31.2	−9.4	8.4 × 10^−5^
420	257.5	277.8	20.3	−5.8	3.0 × 10^−3^
440	275.9	286.0	10.1	−2.8	6.4 × 10^−2^
**460**	294.5	294.2	−0.3	0.1	**1.1**
480	313.2	302.5	−10.7	2.7	1.5 × 10
500	326.6	307.6	−19.0	4.6	9.7 × 10
520	350.7	319.0	−31.6	7.3	1.5 × 10^3^
540	369.5	327.2	−42.2	9.4	1.2 × 10^4^
560	388.3	335.5	−52.8	11.3	8.4 × 10^4^
580	407.1	343.7	−63.4	13.2	5.2 × 10^5^
600	425.8	352.0	−73.8	14.8	2.7 × 10^6^

^a^ The approximation of the ln *K*′-values with the linear equation ln *K*′ = −28.0 × (1000/T) + 61.1 with *R*^2^ = 0.9997.

**Table 14 materials-16-02924-t014:** Gas-phase standard molar Gibbs energies of 2-methyl-8H-indole and 2-methyl-indole and thermodynamic parameters of the dehydrogenation reaction R-IV at *p°* = 1.0 MPa ^a^.

*T*	$\Delta_{f}G_{m}(T)$ _2-Me-8H-Ind_	$\Delta_{f}G_{m}(T)$ _2-Me-Ind_	$\Delta_{r}G_{m}(T)$ _R-IV_	ln *K*′	*K*′
K	kJ·mol^−1^	kJ·mol^−1^	kJ·mol^−1^		
298.15	149.9	230.7	80.8	−23.4	6.9 × 10^−11^
300	151.4	231.3	79.9	−22.8	1.2 × 10^−10^
320	168.4	238.3	69.9	−17.1	3.9 × 10^−8^
340	185.7	246.0	60.2	−12.1	5.6 × 10^−6^
360	203.4	253.8	50.4	−7.6	4.8 × 10^−4^
380	221.2	261.7	40.5	−3.6	2.7 × 10^−2^
400	237.5	268.7	31.2	−0.2	8.4 × 10^−1^
**420**	257.5	277.8	20.3	3.4	**3.0**
440	275.9	286.0	10.1	6.5	6.4 × 10^2^
460	294.5	294.2	−0.3	9.3	1.1 × 10^4^
480	313.2	302.5	−10.7	11.9	1.5 × 10^5^
500	326.6	307.6	−19.0	13.8	9.7 × 10^5^
520	350.7	319.0	−31.6	16.5	1.5 × 10^7^
540	369.5	327.2	−42.2	18.6	1.2 × 10^8^
560	388.3	335.5	−52.8	20.5	8.4 × 10^8^
580	407.1	343.7	−63.4	22.4	5.2 × 10^9^
600	425.8	352.0	−73.8	24.0	2.7 × 10^10^

^a^ The approximation of the ln *K*′-values with the linear equation ln *K*′ = −28.0 × (1000/T) + 70.4 with *R*^2^ = 0.9997.

**Table 15 materials-16-02924-t015:** Gas-phase standard molar Gibbs energies of 2-methyl-8H-indole and 2-methyl-indole and thermodynamic parameters of the dehydrogenation reaction R-IV at *p°* = 2.0 MPa ^a^.

*T*	$\Delta_{f}G_{m}(T)$ _2-Me-8H-Ind_	$\Delta_{f}G_{m}(T)$ _2-Me-Ind_	$\Delta_{r}G_{m}(T)$ _R-IV_	ln *K*′	*K*′
K	kJ·mol^−1^	kJ·mol^−1^	kJ·mol^−1^		
298.15	149.9	230.7	80.8	−20.62	1.1 × 10^−9^
300	151.4	231.3	79.9	−20.03	2.0 × 10^−9^
320	168.4	238.3	69.9	−14.28	6.3 × 10^−7^
340	185.7	246.0	60.2	−9.33	8.9 × 10^−5^
360	203.4	253.8	50.4	−4.86	7.7 × 10^−3^
380	221.2	261.7	40.5	−0.84	4.3 × 10^−1^
**400**	237.5	268.7	31.2	2.60	**1.3**
420	257.5	277.8	20.3	6.17	4.8 × 10^2^
440	275.9	286.0	10.1	9.23	1.0 × 10^4^
460	294.5	294.2	−0.3	12.06	1.7 × 10^5^
480	313.2	302.5	−10.7	14.66	2.3 × 10^6^
500	326.6	307.6	−19.0	16.56	1.6 × 10^7^
520	350.7	319.0	−31.6	19.30	2.4 × 10^8^
540	369.5	327.2	−42.2	21.39	2.0 × 10^9^
560	388.3	335.5	−52.8	23.32	1.3 × 10^10^
580	407.1	343.7	−63.4	25.14	8.3 × 10^10^
600	425.8	352.0	−73.8	26.78	4.3 × 10^11^

^a^ The approximation of the ln *K*′-values with the linear equation ln *K*′ = −28.0 × (1000/T) + 73.1 with *R*^2^ = 0.9997.

**Table 16 materials-16-02924-t016:** Approximation of the temperature dependence of the heat capacity of HL and HR materials by the linear equation $C_{p,m}^{o}=a\cdot ln\left(T\right)+b$.

Compound	*a*	*b*	*R* ^2^
	**Hydrogen-lean** (**HL**)		
indole	151	−738	0.9997
2-Me-H-indole	169	−822	0.9994
indoline	168	−834	0.9993
2-Me-indoline	187	−894	0.9999
	**169** ^a^		
	**Hydrogen-rich** (**HR**)		
H8-indole	229	−1157	0.9980
2Me-H8-indole	250	−1253	0.9982
	**240** ^a^		

^a^ Averaged values recommended for calculations.

## Data Availability

Not applicable.

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
