# Peer review of "Thermodynamic Analysis of Chemical Hydrogen Storage: Energetics of Liquid Organic Hydrogen Carrier Systems Based on Methyl-Substituted Indoles"

_materials, 2023, doi:10.3390/ma16072924_

Round 1

Reviewer 1 Report

Evaluation of the paper named “Thermodynamic analysis of chemical hydrogen storage: energetics of Liquid Organic Hydrogen Carrier systems based on methyl substituted indoles.

Overview

The paper reported the consistent data set for the thermochemical properties of methyl-substituted indoles determined by an approach based on different methods for mutual validation: Experimental measurements utilizing combustion calorimetry and differential scanning calorimetry have been combined with quantum-chemical methods and a group-additivity approach.

Authors confirmed the lower enthalpy of reaction for dehydrogenation in
the five-membered ring and the positive effect of lower enthalpy of reaction by the addition of the nitrogen-heteroatom compared to the corresponding homocyclic compound.
The further author claims that this does not only influence the heat demand for hydrogen release, but also the reaction conditions as it influences the equilibrium constant.

The stated work is interesting and original, nonetheless, numerous problems still can be found. Therefore, I think this paper could be considered for publication after minor revision. 

1.       In Figures 1-3, a, b, c, etc. should be mentioned against each structure of the compounds.

For example.

Structures of hydrogen-lean (HL) counterparts of the LOHC systems: a) 3-methyl-indole, b) 2,3-dimethyl-indole, c) 1,2-dimethyl-indole, and d) 1,3-dimethyl-indole studied in this work.

The same should reflect i.e., in Figures (1-3, page 3) for a clear understanding of the reader.

2.       What is the commercial origin of compounds 3-methyl-indole and 1,2-dimethyl-indole? This should be mentioned in section 2.1.

3.       Compare the values with already reported work ‘’Thermodynamics of Hydrogen Storage: Equilibrium Study of Liquid Organic Hydrogen Carrier System 1-Methylindole/octahydro-1-methylindole” and with

Comprehensive Thermodynamic Study of Alkyl-Cyclohexanes as Liquid Organic Hydrogen Carriers Motifs’’

            What’s new in this?

4.       In table 8, Colum 1 “. compound” should be corrected as “Compound”.

Reviewer 2 Report

In the manuscript " Thermodynamic analysis of chemical hydrogen storage: energetics of Liquid Organic Hydrogen Carrier systems based on methyl substituted indoles " Verevkin et al. presents indoles as the interesting candidates for realizing hydrogen uptake and release by catalytic hydrogenation and dehydrogenation, respectively. A combination of experimental measurements, quantum-chemical methods and a group-additivity approach have been applied to obtain a consistent data set on the enthalpies of formation of different methylated indole derivatives and their hydrogenated counterparts. This manuscript is well-organized and carefully written. It can be accepted after minor revision. The comments are presented as follows:

1. Compared with the solid hydrogen storage materials, the merit of liquid organic hydrogen carriers should be displayed. This article should be cited: Wang Yaxiong, Zhong Shunbin, Sun Fengchun. Research Progress in Vehicular High Mass Density Solid Hydrogen Storage Materials. Chinese Journal of Rare Metals. 2022,46(6):796-812.

2. Among various liquid organic hydrogen carriers, such as formic acid, N-ethyl-carbazole, the merit of Indole derivatives should be discussed. The latest literature about hydrogen storage technologies should be cited, such as Chao Wan, Liu Zhou, Suman Xu, Biyu Jin, Xin Ge, Xing Qian, Lixin Xu, Fengqiu Chen, Xiaoli Zhan, Yongrong Yang, Dangguo Cheng. Defect engineered mesoporous graphitic carbon nitride modified with AgPd nanoparticles for enhanced photocatalytic hydrogen evolution from formic acid, Chemical Engineering Journal, 2022, 429, 132388. Mark D. Allendorf, Vitalie Stavila, Jonathan L. Snider, Matthew Witman, Mark E. Bowden, Kriston Brooks, Ba L. Tran & Tom Autrey. Challenges to developing materials for the transport and storage of hydrogen. Nature Chemistry2022, 14, 1214–1223.

3. In reactant containing benzene ring, less than 3 equivalents of hydrogenation products may be considered and calculated.

Reviewer 3 Report

The draft by Vostrikov et al. describes the thermodynamic modelling of reaction parameters in hydrogenation studies of indole-based materials, especially those substituted with methyl groups. Some minor observation regarding this draft:

-Figure 2: Structure of intermediates….

- Reactions (1)-(3) should be better visualized if accompanied by a figure, where these reactions are put in the context of Hess’ Law.

- The confidence level (page 12) should be described in brief

- The novelty of this study compared to the previous works of the authors: 10.1021/acs.iecr.0c04069 and 10.1016/j.fuel.2022.125764.

- Table S2 from ESI has the entry of 2-pyroline assigned the wrong formula.

Reviewer 4 Report

The paper is a valuable extension of the methods proposed by the same authors for the hydrogenation/dehydrogenation of indole and 2-methylindole to more complex coumpounds belonging to the indole class. The paper is well written and sound. However, some  changes will be useful before publication:

- Please check the numbers in the abstract. It seems that there is a factor 10 with the values reported in section 4 (Table 10). The same seems true also for the numbers reported in the conclusions.

- Figures 1, 2 and 3 could be skipped and the names of the compounds could be reported in Figure 4.

- Table 3: I am really surprised by the large difference of energies between the cis-1,2-dimethyl-(H8)-indole and the trans-1,2-dimethyl-(H8)-indole structures. Please, double check, it seems that they are not the proper structures.

- Use the "." (dot) as decimal separator in Figure 13.

- The authors represent the dehydrogenation of indoles as a two step process, with the release of 1 molecule H2 from the double bond and the second one which correspond to the release of 3 molecules of H2 from the three double bonds. It would be highly interesting to check just in one case if the energies involved in the release from the 3 double bonds are equivalent or they are different if the release of one molecule occurs before the others.

Round 2

Reviewer 4 Report

I appreciate the efforts to improve the manuscript. 

But I have to tell the authors that I still have a problem with the figures of Table 3. In cis-1,2-dimethyl-(H8)-indole, hydrogen was removed both from the 6 and the 5 member rings. In trans-1,2-dimethyl-(H8)-indole hydrogen is removed only from the 6 member ring.

Please, verify again the names and the figures of these two compounds.
